# Brown trout (*Salmo trutta*) originating from warmer streams in Iceland exhibit increased energetic efficiency
Eoin J. O'Gorman [1,7] ✉, Alexia M. González-Ferreras [1,2,7], Penelope S. A. Blyth [3,4], Jamie Coughlan[5],
Jack Hawksley[3], Phil McGinnity[5], Karl P. Phillips[5,6] & Thomas E. Reed [5] ✉

Metabolic rate determines the amount of energy an organism needs to survive, and it is typically
predicted to increase with warming up to an optimum temperature for ectothermic organisms. Once
their metabolic demands have been met, organisms can use the excess energy from feeding for
enhanced growth and reproduction. Experimental evidence suggests that metabolic rate may
increase more with warming than energy intake, which could lead to energetic inefficiency and
population decline. Downregulating metabolic rates or enhancing feeding rates after chronic exposure
to warmer environments could help overcome this problem, but populations and individuals may vary
in their capacity for such change. Here, we experimentally measured the temperature-dependent
metabolic and feeding rates of brown trout (*Salmo trutta*) originating from one cold and two warm
streams in the same geothermally heated catchment, and examined their population genetic structure.
We found a consistent increase in metabolic rate with temperature for all fish, but a stronger increase in
feeding rate with temperature for those originating from warm streams. This resulted in the latter
exhibiting a greater energetic efficiency with increasing temperature than the fish originating from the
cold stream. We detected significant genetic differentiation at neutral markers between the cold and
warm streams, implying limited gene flow across the thermal or geographic gradient, and thus scope
for adaptive divergence. Collectively our results point towards important variation in eco-physiology
within a single catchment that has implications for population persistence in the face of warming.
These results highlight the importance of considering intraspecific variation in predictive models of
biological responses to climate change. They moreover emphasise how energy intake versus
expenditure can be differentially thermally sensitive even at fine spatial scales.

Freshwater ecosystems and the biodiversity they support are among the most endangered globally[1,2]. Freshwaters are particularly vulnerable to climate change because they are exposed to multiple anthropogenic pressures (e.g., water pollution, flow modification, habitat degradation)[3,4], and both water quality and quantity are climate-dependent[5,6]. Consequently, alterations in the magnitude, frequency, duration, timing, and variability of thermal and hydrological events by climate change are affecting biodiversity, ecosystem functioning, and ecosystem services[7]. This is of direct relevance for ectotherms, which constitute the vast majority of freshwater organisms, since their physiology and behaviour are dependent on environmental temperature[8,9].

Temperature is known to influence many biological rates of ectotherms[10], such as metabolism (energy expenditure) and feeding (energy intake), which subsequently determine the energetic efficiency of an organism, i.e., the ratio of energy intake to expenditure. According to the Metabolic Theory of Ecology (MTE), metabolism is primarily driven by

[1]School of Life Sciences, University of Essex, Colchester, UK. [2]IHCantabria - Instituto de Hidráulica Ambiental de la Universidad de Cantabria, Santander, Spain.
[3]Department of Life Sciences, Imperial College London Silwood Park Campus, Berkshire, UK. [4]School of Biosciences, University of Sheffield, Sheffield, UK.
[5]School of Biological, Earth and Environmental Sciences, University College Cork, Distillery Fields, Cork, Ireland. [6]Canadian Rivers Institute, University of New
Brunswick, Fredericton, NB, Canada. [7]These authors contributed equally: Eoin J. O'Gorman, Alexia M. González-Ferreras. ✉e-mail: e.ogorman@essex.ac.uk;
treed@ucc.ie

physiological processes that are subject to the laws of thermodynamics[11]. Thus, the logarithm of metabolic rate generally rises linearly with temperature over the range an organism normally experiences[12,13] with a slope, known as the 'activation energy', of approximately 0.65 eV[11]. Additionally, metabolic rate tends to increase with body mass according to a power law with a log-log allometric slope of approximately 0.75 across all organisms[11,14,15]. Whilst the universality of these values has been questioned, with suggestions of large variation among taxa[16,17], the activation energy is almost always positive and thus global warming is widely predicted to increase the metabolic demands of organisms.

MTE predicts that the feeding rate should rise at the same rate that metabolism increases with temperature[11]. Therefore, as energy expenditure increases exponentially with warming, individual organisms generally need to increase their feeding rates to meet their nutritional demands or face starvation[18,19]. However, experimental evidence suggests a mismatch between the temperature scaling of metabolism and consumption, showing that metabolic rates frequently rise more quickly than feeding rates in warmer environments[18–20]. This could lead to consumers having a reduced energetic efficiency, decreasing their survival, growth, and reproduction[18,21] and leading to unstable population dynamics[22]. Moreover, larger organisms at higher trophic levels are predicted to experience a disproportionate reduction in energetic efficiency, which may have cascading effects on community structure (e.g., changes in food chain length)[23].

Thus, studies are needed to quantify the effects of warming on the energetic efficiency of apex predators due to their role as keystone species and biotic multipliers of climate change impacts[24,25]. However, several previous studies have shown that when energy supply is sufficient, and warming does not exceed the thermal limits of the species, ectotherms can overcome energetic constraints by increasing their energy intake or adapting their feeding to more abundant or energetically valuable resources[26,27]. Alternatively, organisms could downregulate their metabolism following sustained exposure to warmer environments[28,29]. For example, wild fish populations that have been exposed to chronic warming for many generations have been shown to exhibit lower basal metabolic rates than ambient populations, which can contribute to faster growth rates[30,31]. Accordingly, changes in the thermal sensitivity of metabolic and feeding rates that maximise energetic efficiency could stabilise population dynamics in the face of warming[32]. It is thus key to understand how warming may alter both metabolic demand and predator–prey interaction strengths to improve our ability to forecast the effects of global warming on population dynamics, community structure, and food web stability.

Species can adjust to climate change through various morphological, behavioural, physiological, and life history mechanisms to avoid individuals perishing or for local or even global extinctions to occur[33]. These responses include distributional shifts (moving to new areas to remain within their thermal niche), acclimation through phenotypic plasticity (altering their phenotypes as a function of the environment with unchanged genotypes, often as a short-term response within the lifetime of an individual)[34], or genetic adaptation through evolutionary changes (natural selection driving changes in allele frequencies)[35,36]. The scope for such adaptive responses depends on the life history and dispersal traits of the species in relation to habitat, its position within the thermal performance curve, the amount of genetic variation in fitness-related traits, and the magnitude and rate at which its environmental temperature increases[37,38]. Moreover, past eco-evolutionary processes can influence the scope for future adaptation and persistence in the face of environmental change. For example, different populations of the same species may have evolved unique adaptations or differ in the extent to which they are currently adapted to local selective regimes[39], which may render them differentially sensitive to future climate change. Moreover, local populations may experience different rates of warming as broad-scale climate signals get filtered through local habitats. Consequently, it is important to consider how intraspecific variation in climate sensitivity and exposure, as well as overall adaptive capacity, together influence the vulnerability of individual populations and overall metapopulations or species to climate change[40–42].

The ecological and evolutionary responses of apex predators to temperature remain poorly understood. A key reason is that the timescale of most experimental studies is too short to observe evolutionary responses or transgenerational acclimation[43] in these typically long-lived organisms[44], whilst space-for-time studies along latitudinal or altitudinal gradients in nature often suffer from strong confounding factors[45]. Studying intraspecific variation in thermal biology along geothermal gradients at local scales and using apex predators originating from different thermal environments may overcome some of these limitations[30,46]. Previous studies carried out in geothermal systems have suggested that physiological acclimation and/or genetic adaptation to distinct thermal regimes may produce intraspecific variation in thermal sensitivity of metabolism and feeding[27,30–32,47]. However, even though spatial thermal gradients may promote genetic diversity and genetic differentiation across the landscape[48], integration of studies incorporating both functional trait approaches and population genetics in natural systems is scarce. This combined approach is important to elucidate mechanisms through which the predicted effects of global warming may influence the persistence of apex predators.

In this study, we quantify the effects of increasing temperature on the energetic efficiency of brown trout (*Salmo trutta* Linnaeus, 1758) sampled from streams of differing thermal regimes in the Hengill geothermal valley, Iceland. Brown trout is the apex predator and only fish species in the system, and its populations are composed of a high percentage of stationary and low percentage of mobile individuals based on a previous mark-recapture study[27], implying that individuals sampled in a given stream (i.e., living under a particular thermal regime) are likely to have been spawned there, or to have dispersed there early in life and then settled. Our first aim was to quantify the thermal sensitivities of metabolic and feeding rates (and thus energetic efficiency) as a function of the thermal origin of fish. We hypothesised that fish originating from warmer streams should exhibit reduced thermal sensitivity of metabolic rate relative to that of feeding rate across a broad range of temperatures because populations genetically adapted or physiologically acclimated to chronically warmer environments should be optimised to exhibit greater energetic efficiency with warming[32]. Our second aim was to characterise population structure using neutral microsatellite markers, to provide additional context and indirect information on putative patterns of dispersal. We hypothesised that neutral genetic differences would exist between populations from cold and warm streams owing to genetic drift coupled with reduced gene flow, with the latter driven by geographic factors (isolation-by-distance) and temperature differences (isolation-by-environment)[49], or both. Limited gene flow would, in turn, increase the scope for local adaptation with respect to the thermal regime to have evolved in this system. Alternatively, if extensive gene flow occurs among warm and cold streams (panmixia), then any phenotypic differences between fish with respect to source stream (i.e., where they were residing when sampled) could instead reflect phenotypic plasticity, wherein individuals acclimate physiologically to the thermal environment they (or their parents) happen to have dispersed to.

Our experimental design entailed collecting fish from three different source streams (one cold: IS12; two warm: IS1 and IS5) and then transplanting them temporarily to a set of different "experimental" streams in the same catchment, which themselves varied naturally in thermal regimes (Fig. 1). In this way, each source population was artificially exposed to the same thermal gradient (range of experimental temperatures), exploiting natural spatial variation in temperatures in the system and thereby emulating the effects of acute warming in a space-for-time substitution. For logistical and ethical reasons, each fish from a given source stream was transplanted to only a single experimental stream, rather than sequentially moving the same individual to all experimental streams and their associated temperatures (which would introduce order effects and potentially incur excessive stress). Thus, whilst we could not measure within-individual sensitivity of routine metabolic (and feeding) rate to acute warming, we could measure population-level sensitivity. Given that individuals were randomly allocated to experimental streams, any variation in thermal responsiveness among fish originating from the same source stream should

**Fig. 1 | Overview of the Hengill system and experimental approaches. a** Map of the Hengill geothermal system, indicating the streams from which brown trout and experimental prey were collected, and the streams in which experiments were conducted. Note the distance from IS1 to IS12 is approximately 1 km, but the length of the main river is shortened (as indicated by two diagonal lines on the map) for the purposes of viewing the entire system more effectively. Stream codes are the same as those used in previous publications[79,87,88,91]. The stream diagram is adapted from hand-drawn maps used in our previous publications[106,107], and the outline of Iceland was created by using an extracted and edited base outline. **b** An experimental chamber used for in situ metabolic rate measurements, containing a miniDOT logger for monitoring dissolved oxygen concentration and a single individual brown trout. **c** Example of an experimental run for measuring metabolic rates of fish. **d** Example of an experimental run for measuring the feeding rates of fish.

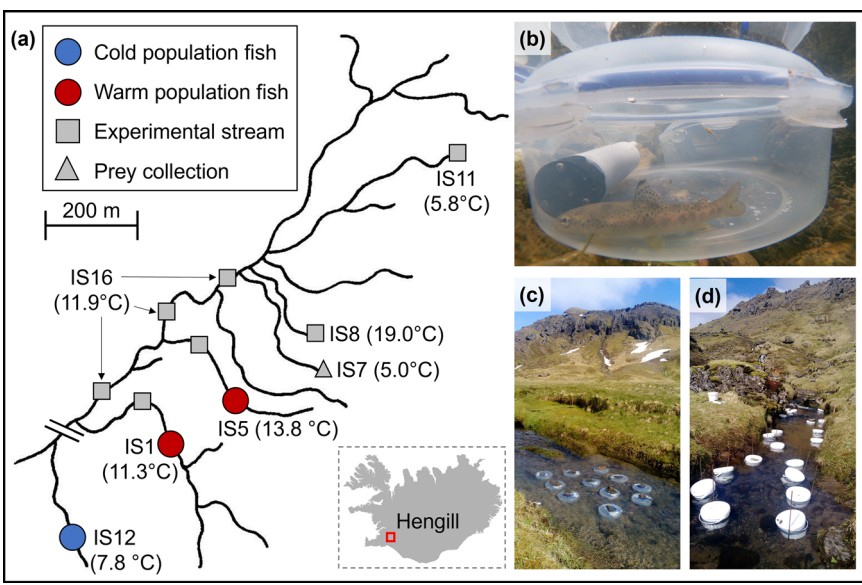

**Fig. 2 | Effects of source-stream thermal regime on metabolic rate.** The optimum model describing variation in the routine metabolic rate of brown trout included main effects of **a** experimental temperature and **b** body mass, but no significant main or interactive effects of source-stream thermal regime (Linear regression: $y = -1.098 + 0.7986 \times \log(M) + 0.3693 \times T{-}T_0/kTT_0$; $F_{2,83} = 86.49$; $p < 0.001$; $r^2 = 0.67$). Thus, a single regression line is fitted in each panel because the source-stream thermal regime (warm v cold) was not included in the optimum model. Note that relationships for each explanatory variable are visualised at the mean value of the other variable. Experimental temperatures correspond to the mean of the values directly measured from the miniDOT logger in each metabolic chamber. Shaded areas are the 95% confidence intervals around the fitted regression lines.

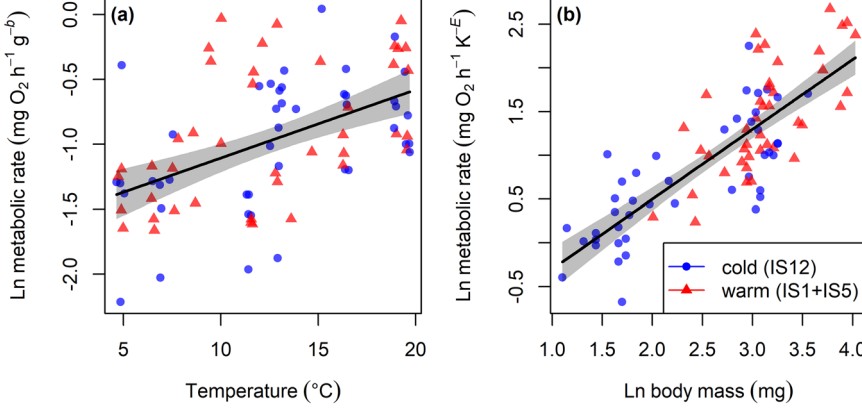

reflect acute effects of temperature on physiology, whilst any differences among fish from different source streams should reflect pre-existing (persistent) variation due to inherited (genetic or epigenetic) factors or prior (early-life) acclimation to developmental temperatures experienced in the home stream.

## Results

### Metabolic rate

The most parsimonious model describing variation in routine (field-measured) metabolic rate of brown trout included only the main effects of experimental temperature and body mass, i.e., source-stream thermal regime had no significant effect on metabolic rate (Supplementary Table 1). There was a significant increase in metabolic rate with both experimental temperature (Linear regression: $F_{1,83} = 27.38$; $p < 0.001$; Fig. 2a) and body mass (Linear regression: $F_{1,83} = 145.6$; $p < 0.001$; Fig. 2b). Similar results were obtained when considering trout collected from the three streams in the system as separate populations, rather than pooling IS1 and IS5 into a single warm population (Supplementary Fig. 1; Supplementary Table 2).

### Feeding rate

The most parsimonious model describing variation in the feeding rate of brown trout on the snail, *Radix balthica*, included main and interactive effects of source-stream thermal regime and experimental temperature, but

no effect of body mass (Supplementary Table 3). There was an increase in feeding rate with increasing temperature for the warm population, but no change in feeding rate with temperature for fish originating from the cold stream (Linear regression, temperature × source-stream term: $F_{1,75} = 5.333$; $p = 0.024$; Fig. 3a). Similarly, the most parsimonious model describing variation in the feeding rate of brown trout on the blackfly larvae, *Simulium vittatum*, included main and interactive effects of source-stream temperature and experimental temperature, but no effect of body mass (Supplementary Table 4). There was an increase in feeding rate with increasing temperature for the warm population, but no change in feeding rate with temperature for the cold population (Linear regression, temperature × source-stream term: $F_{1,84} = 6.859$; $p = 0.010$; Fig. 3b). Similar results were obtained when considering trout collected from the three source streams in the system as separate populations (Supplementary Fig. 2; Supplementary Table 5).

### Energetic efficiency

The most parsimonious model describing variation in the energetic efficiency of brown trout feeding on the snail, *R. balthica*, included a main effect of body mass and main and interactive effects of source-stream thermal regime and experimental temperature (Supplementary Table 6). There was an increase in energetic efficiency with increasing experimental temperature for the warm population, but a decline in energetic efficiency for the cold

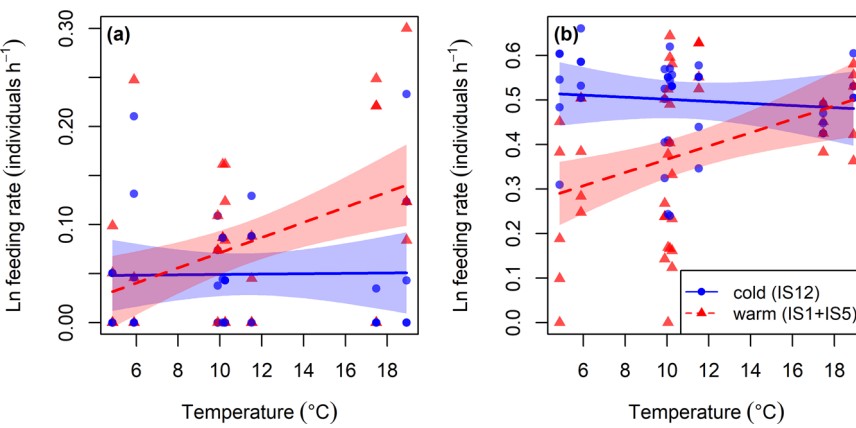

**Fig. 3 | Effects of source-stream thermal regime on feeding rate.** The optimum model describing variation in the feeding rate of brown trout included main and interactive effects of source-stream thermal regime and experimental temperature, but no effects of body mass for both **a** *Radix balthica* (Linear regression: $y = 0.0490 + 0.0013 \times T - T_0/kTT_0 + 0.0234 \times S_{warm} + 0.0528 \times T - T_0/kTT_0 \times S_{warm}$; $F_{3,75} = 5.178$; $p = 0.003$; $r^2 = 0.14$) and **b** *Simulium vittatum* (Linear regression: $y = 0.5011 - 0.0164 \times T - T_0/kTT_0 - 0.1317 \times S_{warm} + 0.1212 \times T - T_0/kTT_0 \times S_{warm}$; $F_{3,84} = 8.69$; $p < 0.001$; $r^2 = 0.21$). Experimental temperatures correspond to the mean value across the period of feeding rate measurements of the stream to which fish were transplanted. Shaded areas are the 95% confidence intervals around the fitted regression lines.

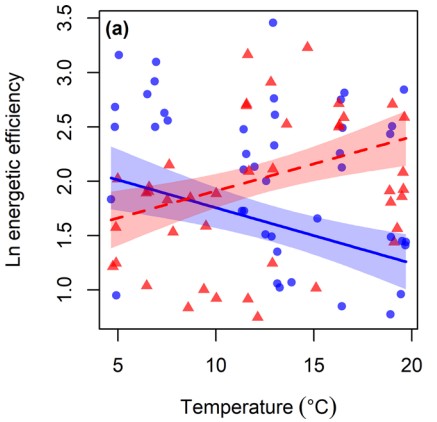

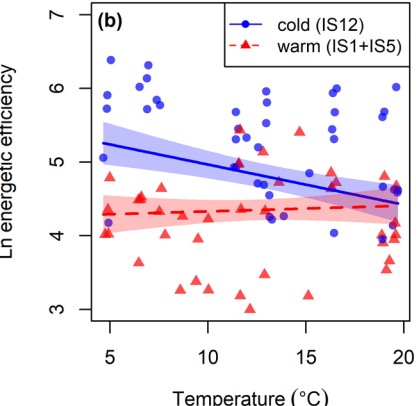

**Fig. 4 | Effects of source-stream thermal regime on energetic efficiency.** The optimum model describing variation in the energetic efficiency of brown trout included a main effect of body mass and main and interactive effects of source-stream thermal regime and experimental temperature for both **a** *Radix balthica* (Linear regression: $y = 3.971 - 0.7528 \times \log(M) - 0.3592 \times T - T_0/kTT_0 + 0.1727 \times S_{warm} + 0.7085 \times T - T_0/kTT_0 \times S_{warm}$; $F_{4,81} = 29.6$; $p < 0.001$; $r^2 = 0.57$) and **b** *Simulium vittatum* (Linear regression: $y = 7.169 - 0.7491 \times \log(M) - 0.3825$ $\times T - T_0/kTT_0 - 0.6251 \times S_{warm} + 0.4369 \times T - T_0/kTT_0 \times S_{warm}$; $F_{4,81} = 55.41$; $p < 0.001$; $r^2 = 0.72$). Experimental temperatures correspond to the mean of the values directly measured from the miniDOT logger in each chamber in the metabolic experiments (in the streams to which fish had been transplanted). Note that energetic efficiency is a dimensionless ratio between metabolic and feeding rates and so does not have units. Shaded areas are the 95% confidence intervals around the fitted regression lines.

population (Linear regression, temperature × source-stream term: $F_{1,81} = 24.96$; $p < 0.001$; Fig. 4a). Similarly, the most parsimonious model describing variation in the energetic efficiency of brown trout feeding on the blackfly larvae, *S. vittatum*, included a main effect of body mass and main and interactive effects of source-stream thermal regime and experimental temperature (Supplementary Table 7). There was an increase in energetic efficiency with increasing temperature for the warm population, but a decline in energetic efficiency for the cold population (Linear regression, temperature × source-stream term: $F_{1,81} = 9.483$; $p = 0.003$; Fig. 4b). Similar results were obtained when considering trout collected from the three source streams in the system as separate populations (Supplementary Fig. 3; Supplementary Table 8).

### Genetic differentiation among streams
Overall, $F_{ST}$ and $D_{EST}$ were 0.012 and 0.002, respectively, indicating weak genetic differentiation among streams in absolute terms. In relative terms, the differentiation was strongest for the IS12-IS1 (cold v warm) and IS12-IS5 (cold v warm) comparisons, and weakest for the IS1-IS5 (warm v warm) comparison (Table 1; see Fig. 1a for a map of the study site). The exact test for genic differentiation was statistically significant for the IS12-IS1 and IS12-IS5 comparisons, but not the IS1-IS5 comparison (Table 1). When pooling IS1 and IS5 (warm source streams) and comparing them against IS12 (cold source stream), the estimates for $F_{ST}$ and $D_{EST}$ were 0.020 and

0.008, respectively, and the exact test was also statistically significant (Table 1). The lower (5%) confidence bounds on $F_{ST}$ and $D_{EST}$ for all pairwise comparisons were always slightly less than zero (Table 1), reflecting the combined effects of weak differentiation and relatively low sample sizes. The negative point estimates for $F_{ST}$ and $D_{EST}$ for the IS1-IS5 (warm-warm) comparison effectively imply zero differentiation. Within-stream genetic diversity is discussed in Supplementary Results.

### Genetic clustering
We retained eight PCs out of a potential 46, representing 58.4% of the original variance. There was a clear separation between IS12 (cold) and IS5 (warm) along the primary DAPC axis, with some overlap between both streams and IS1 (warm; Fig. 5a). DAPC posterior assignments were correct for 20/24 fish from IS12, 10/15 fish from IS1, and 5/8 fish from IS5 (83%, 67% and 60% respectively; 74% overall; null expected accuracy = 39%). All incorrect assignments were either false calls of IS1, or true IS1 fish being called as non-IS1, i.e., there were no instances of IS5 being falsely called IS12 or vice versa, supporting the above interpretation of the DAPC. Among fish belonging to the non-IS1 streams, the error rate resulting in an 'opposite' call was 7.7%, meaning that the vast majority of errors were false calls related to IS1. Combining IS1 and IS5 in line with the genetic differentiation results (Table 1) also showed clear separation into putative "cold" and "warm" populations along the only DAPC axis (Fig. 5b). PCA-based *k*-means

clustering also suggested a best $k$ of 2 (i.e., indicating two genetic populations), which had the lowest BIC value in the tested range and was also the highest $k$ at which clustering was perfectly repeatable. Allocation to the two clusters was 6:2 for IS5, 7:8 for IS1, and 9:15 for IS12, meaning one cluster contained 75%, 47% and 38% of each stream's fish respectively, while the other contained 25%, 53% and 62% (Supplementary Fig. 4), i.e., membership of the first cluster was skewed towards fish from warm source streams, whilst that of the second cluster was skewed towards fish from the cold source stream.

## Discussion

Estimating the thermal sensitivity that determines the balance of energy intake and expenditure of ectotherms is important for understanding and predicting the responses and consequences of populations, communities, and ecosystems to global warming[50,51]. In this study, we hypothesised that fish originating from warmer streams would exhibit reduced thermal sensitivity of metabolic rate relative to feeding rate, increasing their energetic efficiency at higher temperatures, as an adaptive response to a history of chronic exposure to consistently higher average temperatures. In contrast to this hypothesis, we found that brown trout originating from warmer streams did not differ in their thermal sensitivity of metabolic rate compared to trout originating from colder streams, but did exhibit higher thermal sensitivity of both feeding rate and energetic efficiency. This implies pre-existing differences in eco-physiological phenotype among fish from different source streams, which could be explained by their ancestors having experienced consistently different thermal regimes (local adaptation or epigenetic inheritance) or the focal individuals themselves having developed at different temperatures (plasticity or acclimation). Either way, these persistent differences then seem to affect how individuals respond to acute temperature changes within their lifetimes. Our findings indicate that increased energetic efficiency following chronic exposure to warmer environments could be an important physiological response for under-

standing the consequences of global warming at different levels of ecological organisation.

The metabolic rate of brown trout increased with experimental temperature and body mass, leading to higher overall metabolic rates in warmer environments, as predicted for ectotherms by MTE[11]. The thermal sensitivity of metabolic rate ($0.36 \pm 0.14$ eV) was lower than expected by MTE[11], but within the range described by a previous integrated dataset analysis of metabolic rate studies[10,52]. The similar thermal sensitivity across populations from different source-stream thermal regimes is also consistent with some prior studies on fish. For example, there was a similar temperature dependence of metabolic rate in cyprinid fish populations from cooler forest streams (shaded by trees) and warmer farm streams (lacking canopy cover), suggesting that home-stream temperature had no effect on metabolic rate[53]. Likewise, there was no significant difference in the thermal sensitivity of cold- and warm-origin medaka fish (*Oryzias latipes*) populations[54]. These results support the idea that fish exhibit low potential for thermal adaptation in physiologically determined traits such as growth and metabolism[55] (but see ref. [56]). Reduced basal metabolic rates have been observed in stickleback[30] and perch[31] populations originating from warmer thermal regimes, but this does not necessarily contradict our findings, as our study examined routine (field), rather than basal, metabolic rates. Future experiments comparing the thermal sensitivities of metabolic and feeding rates should thus consider basal and maximum metabolic rates, in addition to routine metabolic rates.

According to MTE and its thermodynamic constraints, thermal sensitivity should remain almost constant across organismal traits, and therefore energetic intake and expenditure of an organism should increase with temperature at the same pace for both cold- and warm-origin populations[11,13,57]. This was not the case in our study, however, with feeding rates of trout increasing with temperature only for fish from the warm source streams, and not from the cold source stream. This indicates a divergence in thermal responses across organismal traits, such that fish

**Table 1 | Pairwise comparisons of genetic differentiation among source streams**

| Comparison | $F_{ST}$ ($\theta$) | | | $D_{EST}$ | | | Fisher's exact test | | |
|---|---|---|---|---|---|---|---|---|---|
| | Est | 5% CI | 95% CI | Est | 5% CI | 95% CI | $\chi^2$ | df | *p*-value |
| IS12 (C) v IS1 (W) | 0.014 | −0.011 | 0.046 | 0.004 | −0.014 | 0.032 | 67.1 | 34 | <0.001 |
| IS12 (C) v IS5 (W) | 0.023 | −0.008 | 0.071 | 0.007 | −0.016 | 0.045 | 59.2 | 32 | 0.002 |
| IS1 (W) v IS5 (W) | −0.012 | −0.046 | 0.041 | −0.002 | −0.013 | 0.031 | 20.7 | 34 | 0.965 |
| IS12 (C) v IS1 + IS5 (W) | 0.020 | −0.002 | 0.048 | 0.008 | −0.009 | 0.033 | 87.8 | 34 | <0.001 |

Genetic analysis was performed on the same fish as used in the eco-physiology experiments. Locus *Ssa197* was excluded from the comparison of IS12 v IS5 due to being fixed for the same allele in both populations.
*C* cold source stream (IS12). *W* warm source stream (IS1 and IS5).

**Fig. 5 | Genetic clustering of fish from the study streams.** Discriminant function analysis of principle components (DAPC) shows the separation between: **a** the three source streams from which fish were collected (IS12, IS1, and IS5) with inset eigenvalues highlighting that most of the variation is explained by the primary axis, which has the strongest separation between IS12 (cold) and IS5 (warm); **b** two putative populations (genetic clusters) corresponding to cold (IS12 in blue) and warm (the combination of IS1 and IS5 in red) source thermal regimes, as supported by genetic differentiation among streams (Table 1).

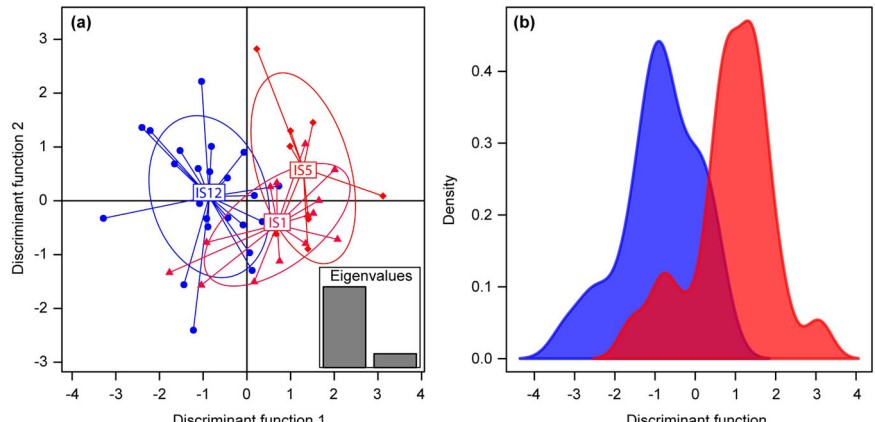

originating from colder regimes may be less able to increase their feeding rate to match their metabolic demands after acute exposure to warmer environments, leading to a reduction in energetic efficiency and thus potential starvation with warming[20]. In contrast, fish originating from warmer regimes may be better able to increase their feeding rate to obtain an increased energetic efficiency with acute warming, highlighting the potential for thermal adaptation to increase population persistence in the face of warming. It must be stressed, however, that we cannot conclude from the current study that trout were genetically adapted to the local thermal regimes of their origin stream. Any persistent differences among these putative populations could instead reflect non-adaptive genetic or purely environmental divergence (phenotypic plasticity). Regardless of the molecular or biochemical mechanisms at play, the differential thermal sensitivity across functional traits may be due to the greater role of cognition in feeding behaviour than metabolism, giving trout greater scope to respond to the needs of a warmer environment after prolonged exposure of individuals or gene pools to high temperatures[58]. Thus, ranking functional traits along a hierarchy of physiological and cognitive contributions may help to explain variation in thermal sensitivity and scope for future adaptive evolutionary or plastic responses across traits. Deviations from MTE have been argued in some studies to be driven by adaptation to local environmental factors[59,60], and our current results suggest that this hypothesis has merit and should be explored by future additional work that combines information on physiology and functional genetics to test more directly for adaptive intraspecific divergence in thermal sensitivities.

Pairwise $F_{ST}$-type metrics (Table 1) indicated significant population-genetic differentiation between IS12 ('cold') and both IS1 and IS5, but not between the latter two warmer streams. DAPC-based analyses – blind and using prior groups – were effective at separating IS12 (coldest) and IS5 (warmest), but not for IS1. Collectively, these results suggest two genetic units in our dataset, corresponding to a "warm cluster" and a "cold cluster" (Fig. 5). These patterns could, in theory, stem entirely from random genetic drift, where allele frequencies in the cold source stream by chance drifted apart from those in the two warm streams under a situation of fully symmetric gene flow. Indeed, effective population sizes are likely to be low in these small streams. A more parsimonious explanation, in our view, however, is that both isolation-by-distance and isolation-by-environment might be going on in this system, and indeed potentially reinforcing one another. That is, gene flow is likely to be constrained between IS12 (cold) and the other two source streams (IS1 and IS5; warm) for both geographic reasons (IS1 an 1S5 are closer to each other than either is to IS12) and because immigrants originating from a different thermal regime might be less successful at passing on their genes in the recipient population, thus amplifying genome-wide divergence driven by genetic drift[49].

With our current study design, we are unable to cleanly separate isolation-by-distance versus isolation-by-adaptation as drivers of neutral genetic divergence among trout originating from warm versus cold streams because temperature differences among the studied streams are confounded with stream distance. Local thermal adaptation correlating with population-genetic structure has been reported in numerous other systems[61,62] (but see refs. 63,64), including for *R. balthica* in distinct geothermal habitats in Iceland[48]. It is thus possible that the contrasting thermal sensitivities that we observed for feeding rate among fish originating from cold versus warm source populations may be due to local adaptation at a genetic level, but this could also reflect non-adaptive genetic divergence, or purely environmental effects on phenotypes (flexible physiological remodelling). Phenotypic plasticity, in turn, could operate within generations (e.g., early-life acclimation[65]) or across generations (e.g., parental effects, epigenetic inheritance[66,67]), and be adaptive, maladaptive or neutral with respect to fitness. Additional experimental work, ideally coupled with functional genetics, would be required to distinguish among these various possibilities, but collectively this highlights the difficulty in anticipating how organisms will respond to warming without incorporating intraspecific variation and its associated eco-evolutionary drivers into predictive frameworks such as MTE.

It is important to note that our goal was not to separate out the relative contributions of phenotypic plasticity and genetic adaptation to any potential differences in the thermal sensitivities of biological rates in fish originating from different streams. Such an undertaking would require removal of the fish from the system and maintenance in a controlled laboratory environment, where second- or third-generation offspring are then used in a reciprocal transplant experiment to control for parental effects and early experience effects on offspring. Our characterisation of neutral population-genetic differences was merely to identify the extent to which the fish in each stream constitute different gene pools (i.e., gene flow is low among streams), and thus the scope for phenotypic divergence to be possibly present. If dispersal is low, then adaptive genetic divergence with respect to thermal regime is more likely to have evolved. Moreover, older fish caught in a particular stream are likely to have been born in that stream, and so any developmental or transgenerational plasticity will then also contribute to any consistent phenotypic differences among streams that underpin differences in biological rates as a response to warming. Studying the physiological or behavioural responses of fish from different population backgrounds in the wild setting of these stream ecosystems is thus a strength of the current study, with disentangling the environmental and genotypic components of any phenotypic divergence a topic for future work.

The temperature effects on foraging were similar for the two prey species used in the experiment. Feeding rate was higher on *S. vittatum* than *R. balthica* for fish from both cold- and warm-origin streams, however, which could be due to the smaller size of the blackfly larvae and thus a need to consume more of them than the larger snails[68,69]. Nevertheless, most of the extra biomass of the snails consists of indigestible shell, meaning the smaller blackfly larvae have a much greater energetic content[70] and thus are a preferred prey item of brown trout[27]. Blackfly larvae also tend to form aggregations[71], which may make it easier to consume multiple individuals, whilst the snails are more mobile, which could increase their ability to evade predation[32,72]. Greater mobility should also make a prey better at evading a predator that is not locally adapted to feeding on it, which may help to explain why fish from the warm source streams outperformed fish from the cold source stream at higher temperatures when feeding on *R. balthica*, but not *S. vittatum*. The same mechanism could also help to mitigate the weaker performance of fish from warm compared to cold source streams at low temperatures (as observed when feeding on *S. vittatum*) if the maladapted predator is more likely to encounter in nature a mobile prey such as *R. balthica*. The effect of warming on trophic interactions thus depends on the temperature dependence of both predators and their prey[73]. Therefore, the next key step is to investigate how prey from different thermal origins respond to warming, instead of the single source stream used here for simplifying the experimental design. This would help to evaluate whether there are contrasting thermal sensitivities for predators and prey, with asymmetry in thermal preferences across trophic levels potentially disrupting population dynamics[74–76]. The effects of chronic exposure of fish to warmer environments on the thermal sensitivity of their feeding rate may not hold across different prey densities. Thus, future studies could also examine if local adaptation or long-term acclimation influences the thermal sensitivity of the functional response of brown trout, which describes the *per capita* feeding rate of a predator as a function of prey density[32,77,78].

Previous research in Hengill showed that brown trout exhibit greater population abundance and biomass as temperature increases due to selective feeding on more energetically valuable prey and a greater nutrient supply, increasing the overall productivity of the warmer streams[27,79]. Our results indicate that greater energetic efficiency, underpinned by either local adaptation or physiological acclimation, could also play an important role in supporting the higher biomass of trout in the warmer streams. This emphasises the value of energetic efficiency for forecasting population abundance or biomass in natural systems[77]. However, it is worth noting that the ratio between energy intake and expenditure was always greater than 1, even for fish originating from the cold stream, indicating that trout had sufficient energy to meet their metabolic demands with warming. In addition to its direct effects on population biomass, increased energetic efficiency

can have indirect effects on ecosystem functioning by altering predator–prey interaction strengths and thus energy flow through the food web[6,74,80]. In this regard, bioenergetics is an important tool for linking organismal physiology with ecosystem processes and forecasting the future effects of warming[81]. However, one of the main challenges to accurate prediction is the inclusion of both ecological and evolutionary processes in ecosystem-level models[82], since forecasts commonly ignore adaptive responses despite their importance for multiple levels of ecological organisation.

The fact that trout may experience long-term adaptation or acclimation to stream temperatures in our study system highlights the potential for salmonids to adjust their biological rates to compensate for warming impacts over the decadal timescales relevant to global climate change. Our results may be more relevant for high latitude ecosystems where organisms often have broader ranges of thermal tolerance and greater physiological flexibility due to the thermal fluctuations they experience, with tropical species that are close to their thermal limits being potentially more vulnerable in this regard[83]. Future studies are also needed to determine whether plastic or evolutionary responses to warming are sufficient to keep pace with or mitigate the effects of global climate change, since thermal adaptation depends on both the extent of environmental variation experienced by an organism and how sensitive they are to that variation[84]. Previous studies have found that faster rates of warming result in a greater metabolic demand for ectotherms compared with gradual rates of warming[85], suggesting that the popular use of altered mean temperature in experimental research may be inappropriate for predicting population-level responses under the increasingly variable temperatures we are experiencing[86]. Thus, it is essential to study how the magnitude, duration, and frequency of extreme thermal events could modify physiological rates and predator–prey interactions in future research.

## Methods
### Study system
The study was conducted from 20th May to 3rd June 2018 in the Hengill valley, Iceland (Fig. 1a), which consists of numerous spring-fed streams flowing into the Hengladalsá river (further details in refs. 27,87–90). Mean annual stream temperatures range from 3 to 20 °C due to differential geo-thermal heating of groundwater through the bedrock[89]. Importantly, the indirect nature of this heating means that physical and chemical char-acteristics are very similar across all streams[87,91], with no significant differ-ence among streams in pH, sulphate, and key nutrients and minerals (Supplementary Fig. 5). A total of 86 brown trout, *Salmo trutta* (65–180 mm fork length), were collected from three streams in the system where they are particularly abundant: 44 from a cold stream (IS12 with a mean annual temperature of 7.8 ± 4.2 standard deviations °C); and 42 from two warm streams (21 from IS1 = 11.3 ± 4.0 °C and 21 from IS5 = 13.8 ± 1.6 °C; see Fig. 1a). IS12 experiences an ambient thermal regime that is typical for Iceland in general, and the other two streams have an environment that is heated by 3.5–6 °C relative to this baseline. Note that we selected fish from two warm streams that were close to each other geographically and in their temperature regimes to ensure we had sufficient numbers for experiments, whilst minimising the impact on these relatively small population sizes. The trout are much more plentiful in IS12, and other cold streams in the system had insufficient numbers, thus a single cold stream sufficed to source fish for the experiments. The gold standard for a reciprocal transplant experiment like this would be to use multiple cold and warm populations, ideally from different river systems, but the small population sizes of brown trout and the uniqueness of the Hengill geothermal system limited the current under-taking to these three streams. We have complied with all relevant ethical regulations for animal use. Electrofishing and handling of brown trout in the experiments were performed in collaboration with the Marine and Fresh-water Research Institute under a permit from the Icelandic Directorate of Fisheries. Icelandic law does not require animal care review for the collection of fish specimens from the wild, and while the research predated the establishment of an ethics committee at the University of Essex, all proce-dures and experiments have been approved by more recent applications to the university's Ethics Review and Management System (ETH2223-0695; ETH2324-1927).

Prior to transplantation to the experimental streams, the trout were maintained in their home streams in cylindrical white plastic arenas (250 mm diameter, 300 mm height). Each arena contained two holes situ-ated directly opposite each other (70 mm diameter, 10 mm from the base), which were covered by 500 μm mesh to allow in-flow of oxygenated stream water, whilst preventing in- and out-flow of potential macroinvertebrate prey. The arenas were partially submerged in the streams such that half of the interior was filled with water, leaving the trout access to the surface as in the natural system. Metal rebars were taped to the arenas and hammered into the stream bed to hold them in place, and a rock was placed inside the arena and on top of the lid to weigh it down. The fish were starved for 48 h prior to experiments to standardise their hunger levels, whilst preventing metabolic down-regulation due to extended starvation[20,92].

### Metabolic rate experiments
Oxygen consumption rates were used as a proxy for metabolic rates[93] and were measured in situ in five streams in the system: IS1, IS5, IS8, IS11, and IS16. Here, these experimental streams acted as a sort of natural laboratory, whereby we could conduct metabolic rate experiments at different tem-peratures (in the experimental streams to which fish from the three source streams were transplanted) without needing to bring fish back to the laboratory to do so in an artificial setting under temperature-controlled conditions. These experimental streams were chosen to best span the range of available temperatures during sampling (4.6–19.7 °C). Note that although fish from IS12 (cold source stream) were included as one of the three study populations, IS12 itself was not included as an experimental stream because it is much further from the others, so it would have involved transporting fish over greater distances and was not needed to evenly span the tem-perature gradient. Static respirometry experiments were conducted in 7.2 L circular plastic chambers (LocknLock, South Korea) with a lid that could be opened and closed to create an airtight seal (Fig. 1b), following an estab-lished methodology[29]. Before placement in the streams, each chamber was fully submerged in a 50 L plastic container that had been filled with water from the experimental stream, filtered through a 250 μm sieve to remove small organisms or plant matter that may affect the level of background respiration[94]. A single brown trout individual from a given source stream was placed inside each chamber along with a MiniDOT logger (PME, USA) to measure dissolved oxygen concentrations and water temperature every minute. The chamber was then sealed underwater to avoid any air bubbles that may interfere with dissolved oxygen readings and completely sub-merged in the experimental stream, with a rock on the lid to secure it to the stream bed (Fig. 1c). Up to ten chambers containing fish were placed in the experimental stream for each block of metabolic experiments, whilst an extra chamber without any fish was included as a control for measuring background respiration. This typically involved five fish from the cold source stream (IS12) and five fish from the warm source streams (e.g., two from IS1 and three from IS5, or vice versa). Occasionally, an experimental run had fewer than ten fish, but we ensured a balanced number of cold and warm-origin fish were used in these cases. Note that fish were transported in buckets from their stream of origin and placed in the experimental stream for 30–60 min prior to being introduced into the metabolic chambers to help equalise temperature and avoid a stress response.

The experiments ran for at least 2.5 h, after which time the fish was removed, its body length was measured, and it was subsequently used in a feeding rate experiment (see below). Oxygen depletion was noticeably linear and relatively stable (see Supplementary Fig. 6), suggesting that the fish experienced minimal stress during the experiments. Nevertheless, the first half hour of data was excluded as an acclimation period for the fish to overcome any effects of handling and adjust to their new environment. To ensure a consistent duration across all experiments, we analysed the sub-sequent 2 h of data using a linear regression of oxygen consumption over time to obtain the oxygen consumption rate [mg $O_2$ $L^{-1}$ $h^{-1}$], which was corrected to a whole organismal rate [mg $O_2$ $h^{-1}$] by multiplying by the

volume of the chamber minus the volume of the fish (assuming a density of 1000 kg m$^{-3}$). To account for background respiration, we subtracted the slope of the control data for the corresponding time period. We excluded any data where the final rate had $r^2 < 0.8$ (5 of 91 experiments). Experiments typically took place in the morning to mid-afternoon when the fish were quite active in the chambers, and so the measurements can be considered an active field metabolic rate.

The effect of source-stream temperature (i.e., whether the fish came from putatively cold- or warm-adapted populations) on the size- and temperature-dependence of metabolic rate, $I$, was determined from the natural logarithm of the following equation:

$$I = I_0 M^b e^{E \frac{T-T_0}{kTT_0}} S \quad (1)$$

where $I_0$ is the metabolic rate at $T_0$, $M$ [mg] is the body mass of an individual fish, $b$ is an allometric exponent, $E$ [eV] is the activation energy (or slope of the log-linear scaling with Arrhenius temperature), $k$ [$8.618 \times 10^{-5}$ eV K$^{-1}$] is the Boltzmann constant, $T$ [K] is the absolute experimental temperature (i.e., the recorded temperature in the experimental stream at the time of each metabolic rate measurement), $T_0$ [283.15 K] is the normalisation temperature (approximately the mean temperature experienced by fish across the experimental streams), and $S$ is a categorical predictor for source-stream temperature (with two levels: cold or warm). We explored all combinations of the main and interactive effects in Eq. 1 using linear regression analysis and chose the most parsimonious model using Akaike Information Criterion corrected for small sample size (AICc).

## Feeding rate experiments

Following each metabolism experiment, the fish were immediately used in a feeding rate experiment conducted in the same experimental stream. Before each experiment, 200 *R. balthica* and 200 *S. vittatum* were hand-collected from an independent stream (IS7). Both taxa are naturally present in the streams and are also well represented in the diet of trout in IS1, IS5, and IS12[91]. Twenty individuals of a particular taxon were added to each of twenty cylindrical arenas (identical to those used to store fish before the metabolic experiments), with ten arenas containing snails and ten containing blackfly larvae. The arenas were secured in the experimental stream with metal rebars and rocks (Fig. 1d). An individual trout was then added to each arena, ensuring that five arenas for each prey taxon contained trout from the cold stream and five contained trout from the warm streams. The experiments ran for approximately 24 h, after which time the remaining prey individuals were counted, and the fish were released back into their home streams. Adipose fins were removed for genetic analysis prior to release, thus also ensuring that no fish would be reused in any further experiments. The feeding rate of fish, $F$ [individuals h$^{-1}$], was calculated as:

$$F = \frac{N_e}{t} \quad (2)$$

where $N_e$ is the number of prey eaten by the fish in an arena and $t$ is the duration of the experiment in hours. We analysed the experimental data with a linear regression obtained from the natural logarithm of Eq. 1, after substituting $F$ for $I$.

## Energetic efficiency

To determine the potential impact of source-stream temperature on the energetic constraints of brown trout evaluated across different experimental temperatures, we calculated a dimensionless energetic efficiency, $y$, as the ratio of feeding rate to metabolic rate:

$$y = \frac{\omega F}{I} \quad (3)$$

Here, $I$ is the actual metabolic rate for each fish as measured in the metabolic rate experiments, whereas $F$ is the model-predicted feeding rate

for a fish of the same size at the exact temperature that the metabolic rate experiment was conducted. This approach enabled us to calculate an energetic efficiency for both prey taxa in energetic equivalents [J h$^{-1}$]. Oxygen consumption rates were converted to energetic equivalents using the density of $O_2$ (1.429 g L$^{-1}$) and a standard energy conversion (1 ml $O_2$ = 20.1 J)[95]. Feeding rates were converted to energetic equivalents using the average body mass of *R. balthica* and *S. vittatum* from IS7[79] and the caloric content of Gastropoda and *Simulium* spp. from an established database of energetic equivalents[70]. Finally, $\omega$ is the unitless assimilation efficiency, which was estimated using an established temperature dependence[96]:

$$\omega = \frac{\omega_0 e^{E_\omega \frac{T-T_0^*}{kTT_0^*}}}{1 + \omega_0 e^{E_\omega \frac{T-T_0^*}{kTT_0^*}}} \quad (4)$$

Here, $\omega_0$ [$e^{2.266}$] is the intercept of the linearised version of Eq. 4 at $T_0^*$ [293.15 K] and $E_\omega$ [0.164 eV] is the activation energy for carnivorous invertebrates[96]. Note that values of $y \geq 1$ in Eq. 3 indicate that the feeding rate of the predator is sufficient to meet its metabolic demands, while values of $y < 1$ suggest that the predator is energetically constrained.

## Genetic differentiation among streams

At the end of all experiments, fin clips were taken for population genetics and preserved in 96% ethanol, with a total of 17 microsatellites genotyped (Supplementary Table 9). Weir and Cockerham's equivalent ($\theta$) of Wright's $F_{ST}$ statistic[97] was estimated for each pairwise comparison between the streams (i.e., IS12-IS1, IS12-IS5, IS1-IS5) using the 'fastDivPart' function in the 'diveRsity' package in R[98]. Lower or higher values of $\theta$ (which range from 0 to 1) indicate lesser or greater divergence, respectively. In addition, we calculated the diversity partitioning $D_{EST}$ metric[99], which is less distorted by the high heterozygosity of microsatellites[100]. Bootstrapping (10,000 replicates) was used to estimate bias-corrected confidence intervals on both $\theta$ and $D_{EST}$[98]. To further test for genetic differentiation among streams, exact conditional contingency-table tests for genic (allele frequency) differentiation were performed using the Genepop software[101], implemented using the 'test_diff' function in the 'genepop' package in R[102], with 10,000 dememorisations, 100 batches, and 5000 iterations per batch. The above analyses were then repeated after merging IS1 and IS5 into a single putative "population", and contrasting this against the IS12 stream. IS1 and IS5 have a more similar temperature profile and are geographically closer to each other (in distance terms) than either is to IS12. Thus, the scope for (effective) gene flow between IS1 and IS5 might be higher than between either stream and IS12, such that they potentially behave more like a single population in genetic (but not necessarily demographic) terms[103]. Note that streams are part of the same river network, meaning that across the isolation by distance template, there are no physical longitudinal barriers or confounding environmental factors other than temperature[87,91]. Quantification of within-stream genetic diversity is described in Supplementary Methods.

## Genetic clustering

We used discriminant function analysis of principal components (DAPC) to assess the degree to which genotypes could be used to assign individuals to their streams of origin, implemented *via* the 'adegenet' package in R[104]. We performed an initial DAPC using the 'dapc' function on centred, unscaled genotype data, using the prior groupings, two discriminant functions, and the first $n$ PCs to account for 95% of PCA variance. To select the final number of PCs, we applied the 'optim.a.score' and 'xvalDapc' optimisation functions, visually inspected the outputs, and retained the higher of the two values. We then performed a final DAPC with this number of PCs and inspected the posterior assignments. To provide an indication of the generalisability of DAPC-based assignments, we performed 10,000 repeats of custom cross-

**Article**

validation, in which we removed a randomly selected 1/8 individuals per prior group, trained a DAPC on the remaining 7/8, and attempted to predict the groups of the removed individuals. To examine genetic clustering and as a control for prior-informed DAPC, we performed blind, *k*-means clustering on the PCA of the genotypes using the '*find.clusters*' function. We performed 20 repeats of a *k* range of 1–7, and visually inspected Bayesian Information Criterion (BIC) values to determine the best *k*.

### Statistics and reproducibility

We performed linear regression analyses with the natural logarithm of metabolic rate, feeding rate, or energetic efficiency as the response variable (i.e., outcome measures) and the natural logarithm of body mass (continuous variable), Arrhenius temperature (continuous variable), and source-stream temperature (categorical with two levels: cold or warm) as explanatory variables. We explored all combinations of the main and interactive effects and chose the most parsimonious model using AICc. Note that fish from the cold stream can be thought of as the control group, with fish from the warm streams experiencing an elevated thermal regime compared to ambient stream temperatures in Iceland. In all cases, replicates were experiments conducted on new individuals with no reusing of fish or invertebrates in any experiment. No sample-size calculation was performed, but the number of usable experiments conducted for metabolic rate ($n = 86$), feeding rate on *R. balthica* ($n = 79$), and feeding rate on *S. vittatum* ($n = 88$) were all more than sufficient to ensure a large enough sample size for linear regression analysis (recommended minimum $n = 30$). All trout less than 65 mm were deemed too small for the metabolic rate experiments and thus released, but we otherwise collected every fish we sampled from electrofishing for use in the experiments. This ensures that the research sample is representative of the typical size, age, and sex structure in each stream. Similarly, we collected the first 200 *R. balthica* and *S. vittatum* we could find in IS7 for use in the feeding rate experiments, and thus these research samples are also representative of the typical size, age, and sex structure of each macroinvertebrate species in the stream. Model validation was conducted on all analyses, and the assumptions of normality, homogeneity, and independence of residuals were met in all cases.

For the metabolic rate experiments, we excluded any data where the linear regression of dissolved oxygen data during the experimental period had an $r^2$ value < 0.8, which occurred in 5 out of 91 experiments (see Supplementary Fig. 6). No data were excluded from the feeding rate experiments. There are few fish in the system, so there was no random allocation to treatments; however, we ensured that a balanced number of cold-origin and warm-origin fish were used in each experimental run and that fish from these two treatment groups were interspersed among experimental arenas rather than grouping all individuals from a particular treatment group at one location in an experimental stream. Samples were processed according to unique codes that were only translated into treatment identifiers once all the data were collected. The experiments were a huge undertaking in a remote Icelandic valley, so no attempt has been made to repeat them or check the reproducibility of the findings.

### Reporting summary

Further information on research design is available in the Nature Portfolio Reporting Summary linked to this article.

### Data availability

The data that support the findings of this study are available from the University of Essex Research Data Repository[105] at https://doi.org/10.5526/ERDR-00000243. Numerical source data underpinning the figures can be found in Supplementary Data. All other data are available from the corresponding authors on reasonable request.

### Code availability

The code that supports the findings of this study is available from the University of Essex Research Data Repository[105] at https://doi.org/10.5526/ERDR-00000243. All analyses were conducted in R v4.0.2.

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

## Acknowledgements

We thank Gísli Már Gíslason, Jón S. Ólafsson, Guðni Guðbergsson and Sigríður Ásgeirsdóttir for providing research support and facilities. We acknowledge the funding support of NERC (NE/L011840/1, NE/M020843/1), the Royal Society (RG140601), Imperial College London, the Government of Cantabria through the Fénix Programme, grant RYC2023-045780-I funded by MICIU/AEI/10.13039/501100011033 and ESF +, the European Research Council (ERC Starting Grant 639192), and Science Foundation Ireland (SFI/15/IA/3028).

## Author contributions

E.J.O. secured funding and designed the study. E.J.O., P.S.A.B., and J.H. conducted the fieldwork. E.J.O. and A.M.G.F. analysed the data. J.C., K.P.P., P.M., and T.E.R. conducted the genetics analyses. E.J.O. and A.M.G.F. wrote the first draft of the paper. All authors edited the paper.

## Competing interests

The authors declare no competing interests. E.J.O. is an Editorial Board Member for Communications Biology, but was not involved in the editorial review of, nor the decision to publish this article.
