## [Transparent Peer Review file · Communications Biology]

Brown trout (*Salmo trutta*) originating from warmer streams in Iceland exhibit increased energetic efficiency

Corresponding Author: Dr Eoin O'Gorman

Version 0:

Reviewer comments:

Reviewer #1

(Remarks to the Author)

This paper, evaluating the potential for differential responses in brown trout to temperature exposure based on origin in a cold or warm streams, is of considerable interest for our understanding of thermal influences on physiology, behavior and ecology, as well as the ability for populations from different thermal environments to respond and keep pace with climate change. The manuscript is beautifully written with nice grounding in the literature and theory, well done. Conclusions are generally reasonably caveated based on the limitations of the study (save for the overly strong title, see below), and the authors thoughtfully pointed to where future work could deepen our understanding of the facets that were more limited in providing conclusive evidence given their design; demonstration of genetic distances is a nice foundation. Analyses are sufficient and data clear – but study design needs more clarification/detail within the manuscript. I have limited comments as generally, this is a solid paper that, with appropriate attention to the needs below, will be a nice addition to the literature.

Title – I think the title is too strong given the lack of population-level replication in the study, e.g. multiple populations each from a cold and warm environment (which also would help address the confounding between geographic and genetic difference). This is what it is, given the system available, but the conclusions need to match the bounds of the design. Something along the lines of “Increased energetic efficiency observed in brown trout from warm streams versus a cold stream in a northern latitude system” would be more appropriate than the strong causative conclusion conveyed in the current title.

The journal may limit figure space, but it would be helpful, if at all possible, to include the map and experimental approaches Figure (S1) in the actual paper, in part as the study design is unclear (to me, at least, see below). The figure legend text describing the visually indicated break in distance between IS12 and IS1 is important but confusing as there is no distance legend given to then convey the comparative distance between IS1 and IS5. Suggest adding a distance legend.

L375 if energy expenditure/intake is greater than 1, doesn't this mean insufficient energy (<1) to meet demands – i.e. should this ratio be flipped (intake/energy >1)?

L429 – first mention of microsatellites as marker used. State in L424 that ___# of microsatellites were genotyped.

Experimental design needs clearer description. L461 made me wonder why IS12 wasn't included in the experimental locales; assumedly it's because these other sites spanned the broad variation in temperatures you were targeting in your experiment (?), so state that. It would also be good if you could indicate, or color code with a legend, the temperatures at each site on the map. Finally, I don't understand how the fish from the cold stream and warm streams were allocated to the experimental design: L 473 said “up to 10 fish” – does this mean 3 from each focal population + the 1 (empty) control at each site? And what happened when fewer fish were used – from which populations? This all just needs a bit more thought given to the narrative.

L508 IS7 not on map – add.

Reviewer #2

(Remarks to the Author)

SUMMARY

This is an interesting paper taking advantage of a study system of cold and warm streams in Iceland that are at contrasting temperatures due to geothermal warming despite their spatial proximity. The findings are interesting (although I have some

concerns about their validity) and the manuscript is mostly well-written. I do have some concerns regarding the methodological and statistical approaches, as well as some suggestions for improving clarity, the presentation of the results, and various other aspects of the manuscript, as detailed below.

MAJOR COMMENTS

- 1) There is a potential issue with the approach in that only one cold stream and two warm streams were used. It is therefore difficult to attribute the different pattern in the “cold fish” to the temperature itself – it is hard to establish a causal relationship without comparing multiple cold and multiple warm populations. Can you eliminate the possibility that there is some other abiotic or biotic factor that differs between these cold and warm streams that could have led to these physiological differences?
- 2) I question the appropriateness of the statistical approach – the authors chose to merge the data for the two warm streams. However, related to the point above, I would argue that it is crucial to show whether or not the two separate warm streams both exhibit the same pattern as each other. If this pattern is driven by only one of the warm streams, that calls into question the conclusion that the observed increased energetic efficiency is due to warming as opposed to some other attribute of the environment in one of these warm streams. Indeed, in Figure S5a, it doesn't look like there is any evidence for increased energetic efficiency in the IS5 population – this seems to only be the case for IS1. Statistical results need to be provided for the main effects and interaction for the relevant supplementary figures for your results to be convincing (in fact, in my opinion, these supplementary figures should be in the main text instead). When looking at these supplementary figures, it also becomes clear that the sample size for IS1 is actually quite small. All sample sizes should be stated clearly in the main text.
- 3) Another concern I have is that the authors appear to have omitted mentioning relevant literature, perhaps in an effort to overstate the novelty of this work. They do not cite studies that seem directly relevant to this topic, including some on the metabolic rate of other species in Iceland that are exposed to chronic warming in geothermally heated freshwater systems (e.g. threespine sticklebacks, Pilakouta et al. 2020 *Functional Ecology*) or species in other locations where there is exposure to chronic warming (e.g. European perch, Sandblom et al. 2016 *Nature Communications*).
- 4) I believe the presentation of the results could be significantly improved.
 - a) For example, why are three stream names IS12, IS1, and IS5? Can you use something more intuitive for presenting the results for these different streams? On L461, you then mention IS1, IS5, IS8, IS11, and IS16, which are designated as “experimental streams” in Figure S1, but it is unclear what this means.
 - b) The use of colour is inconsistent and confusing. IS1 is shown in orange in Figure 1a but then it's merged with IS5 and shown in red in Figure 1b.
 - c) Figure 2: The caption should provide some information on which populations/streams the “cold” and “warm” correspond to. Also, why aren't there separate regression lines shown for cold and warm in these plots? Even if “source stream temperature” doesn't have a significant effect on metabolic rate, it would be useful to show these separate lines on the plots.
- 5) The Discussion is very focused on brown trout and specifically the Hengill study system with limited effort to broaden the scope and relate this to relevant literature. I suggest reframing some of the Discussion to try to place your results in a broader context.

MINOR COMMENTS

- Consider rephrasing the title: “Chronic exposure to a warmer environment increases energetic efficiency of brown trout
- L27-28: This statement is not specific to predatory organisms, right? I suggest rewording.
- L27-30: A bit of a jump in logic between sentences here – why does warming lead to an increase in metabolic rates? Need to specify if talking about ectotherms.
- L29: “However” doesn't fit in the middle of the sentence.
- L33-34: Specify how many cold and warm streams you sampled.
- L92: Avoid using / as it is too informal for a scientific paper.
- L144-160: This feels really out of place as the last paragraph in the Introduction. In my opinion, it would be more appropriate to move this to the Discussion. This would also help shorten the Introduction, which I found a bit too long.
- I suggest avoiding abbreviations such as MTE which require the reader to remember what these refer to, but it is ultimately up to the authors to decide whether to keep this.

Version 1:

Reviewer comments:

Reviewer #1

(Remarks to the Author)

I reviewed the earlier draft of this manuscript which remains well-written and elegantly couched in the literature. I apologize if my original review did not sufficiently emphasize concern about the overly strong conclusions given the minimal study design (which is not by fault of the authors, just a constraint of the setting, with insufficient replication and very small sample sizes), but I'm afraid those concerns still stand. This paper provides a well-crafted theoretical framing around a question of true importance, but one for which the study limitations simply do not align with that lofty setting or enable such firm conclusions. These concerns are elevated at this point given the high caliber of this particular journal, which notes in its guidelines “There should be a discernible reason why the work deserves the visibility of publication in a Nature Portfolio journal rather than the best of the specialist journals.” This is of course an editorial decision, but my recommendation would be that the paper merits publication after further (relatively easy) strengthening of its narrative description of the study design and caveated interpretation of the results (there are slight improvements in this revision but still work to be done in my

opinion) – but in a different, more specialized journal.

Mainly, I feel the conclusions are still too strong given the lack of replication. Also, the expectations for interpreting the genetic data are not appropriately caveated (except for in the discussion) given the confounding influence of isolation by distance and small population sizes on genetic outcomes (I should have commented on this more deeply in my original review so again, I apologize for the rather light input originally).

Given the organization of this journal (introductions, results, discussion, methods), my comments are a bit interspersed, but I hope they are helpful in a next revision for hopeful publication in another journal.

I appreciate the revision of the title to narrow the findings to a given system, but it is still too causative and not appropriately caveated given the limited study design – it should emphasize that increased energetic efficiency was observed (correlated) in this system, not caused by exposure to warmer temperatures.

138: The authors present nice coverage of the limitations of their genetic data in the Discussion, but the set up of genetic outcomes “as a function of the thermal origin” in this hypothesis is one thing that gives me greater pause on 2nd review as, especially with the lack of replication, there is no way to separate the influence of isolation by temperature from isolation by distance (as noted in the Discussion). Additionally, a point not covered in the paper at all is that in small populations like with assumedly small effective sizes, the degree of genetic differentiation observed between the cold and warm streams (without replication) could be entirely explainable by drift as this influence is generally greater in small populations than selection.

164: Even though 2 clusters may be merited, all of these results do not suggest strong affinity for the two clusters by stream thermal profile (i.e., a cold vs hot cluster). They are generally as expected with isolation by distance, with IS1 falling genetically in between the other two streams in line with its being between them geographically.

261: Use of the term “chronic exposure” made me think this was referring to (longer-term) experimental exposure, but you mean this based as a descriptor of where the fish originated – clarify this: “chronic exposure due to originating in a population from a warmer stream”

L283: this specific statement about perch and sticklebacks (“the reduced metabolic rates observed”) makes it seem like you’ve brought this study up before, rather than that you are extending your general discussion based on varied literature including for these fish. Also, the use of “following chronic exposure” here, rather than “for fish originating from populations from different thermal regimes” begs my point above on the use of “chronic exposure”.

293: These kind of statements and the strong conclusions – still – from your title are making me evaluate this paper more critically. This is not founded based on this minimal study design without replication: “This indicates a divergence in thermal responses across organismal traits”

312: “Collectively, these results suggest two genetic units in our dataset...” – this is true, but the following statement is not grounded given the potentially confounding effect of IBD and drift – “...which may be adapted to a normal ambient stream environment for Iceland and one which is heated by 3.5-6 °C”. There is a good, valid discussion of this following shortly, L 321, that could be better integrated/caveated in these earlier firm statements.

L364: This idea (that increasing energetic efficiency may contribute to this observation) is interesting and founded as it is posed. It is appropriately caveated with “could”, but note the following statement is not appropriately caveated based on your results: “underpinned by a genetic component that favours warm populations”

L378: This section appropriately grounds/caveats your interpretation of/goals for the genetic evaluation, in simply gaging if the “scope for phenotypic divergence to be possibly present”. Need to do a better job of adhering to this interpretation in presenting your hypotheses and results.

L387: Related to adaptive divergence, given how small you indicated these populations likely are, their effective population sizes may be too low to allow for adaptive divergence. Rather, this may be more likely to be a drift-dominated system, genetically. Add discussion of this.

L415: I am realizing on this 2nd review that the overarching study manipulation/design was never introduced before going into details. I think the experimental vs study streams aspect to the design needs a clearer, up front narrative explanation that consistently distinguishes *fish* from study populations from different thermal environments and their use in a study design vs the *thermal habitats* of streams used for the in-situ experiment. This description as is gets very into the weeds quickly, and essential elements of this approach are presented piecemeal (L420, L497) without setting up the broader vision. It would benefit from a brief overview description along the lines of: “Our study anchored around three populations of brown trout from different thermal habitats; our experiment consisted of using fish from these 3 study streams in a designed assessment of their metabolic and feeding responses while temporarily exposed to different temperatures in several test streams (or stream reaches of their origin stream) representing the broad thermal variation observed in the system”. This will provide the overview above the subheadings, as a basic study design (for transfer of fish from study populations to thermal experimental streams) applicable to all the manipulations (oxygen consumption, feeding rate, etc). Perhaps describe this under a subheading of General study design.

433: Add “prior to translocation to experimental streams for study, trout were maintained...”

L453: I suggest presenting the genetic methods at the end, as it is confusing here without the clearer study design laid out – and you note the collections were done after the experiments. How you treated fish from these populations (by merging IS1 and 5) given the genetic outcomes can be presented more up-front in the results, but I think should come last in the methods.

473: Suggest strengthening the recognition of the distance/temperature conundrum by removing the word “with” after “network,” and adding “meaning that across the isolation by distance template there are” in this phrase: “...the same river network, with no physical longitudinal barriers or confounding environmental factors other than temperature”. Again, it is nicely stated in Discussion, L260: “consequence of isolation-by-distance and/or isolation-by environment, which was supported by our genetic results” but this limitation needs to be presented here given that distinguishing these two is important for your ultimate conclusions and yet you cannot distinguish them at all.

493: Try to strengthen the clarity around study fish vs experimental streams wherever you can, here, e.g., by adding after “measured” the descriptor “for study stream fish in the experimental design carried out”

498: Add further clarity along these lines: “Note that, though fish from IS12 were included as a study population, IS12 itself was not included as an experimental stream...”

506: Similarly, revise to read “A single brown trout individual from a given study population was placed inside...”

511: Similarly, revise to read “Up to ten chambers containing fish were placed in each thermally-targeted experimental stream for each experimental run,…”

512: This may be somewhat redundant with the comment above but good added detail of the type that is more generally needed (first tangible mention of study fish being used across this design). I would move it, and the following sentence (Note that…) up as part of a general description of the overall experimental manipulation/design before the description of how you did the measurements: “This typically involved five fish from the cold stream (IS12) and five fish from the warm streams (e.g. two from IS1 and three from IS5, or vice versa).” Then you could follow with the “Note that” statement about not including IS12 as an experimental stream, followed by the other “Note that” statement, edited as needed for transition, mentioning the acclimatization of fish in the experimental streams before measurements were taken.

547: Revise to read “in the same thermally-targeted experimental stream”

564: Note to editor, this is not at all my realm of expertise so I am not positioned to evaluate the sufficiency of this approach.

565: This is a good descriptor the type of which could, aligning with my general recommendation for more consistent clarity on design, be included to ground your hypothesis for each subheading/experiment; but suggest revising further to read: “To determine the potential impact of source-stream temperature on the energetic constraints of brown trout evaluated across different experimental temperatures…”

Version 2:

Reviewer comments:

Reviewer #1

(Remarks to the Author)

I reviewed this paper twice previously and commend the authors for addressing my comments and making considerable improvements. I think it is much clearer and hope they agree. I do still feel the study design lacks important replication - and made two comments below I think it would be useful for them to address more clearly in the paper - but at this point will leave it to the editor to decide if what is, effectively, a paired comparison merits publication in this high-level journal. For my part, thanks for the effort to revise this well-presented paper.

L482. This may be true in terms of getting enough fish for sample size but is not a reason to select a single source stream and undermines the importance of replication in design. Assumedly IS7 did not support sufficient fish for use as another, geographically close, cold stream? Can you specify this (similar to next comment)?

L 500 IS8 and 11: Similarly, why were these additional cold and warm streams - at least the cold (11) - not included as study pops? The reader assumes you could have done the transplants into experimental streams that were also source streams, so please detail why this opportunity wasn't taken advantage of (not enough or any fish in these streams, e.g.).

made.

*** Note that original reviewers' comments are in bold. Our responses are numbered sequentially, written in plain font, and contain quotations to the revised text in italics with line numbers corresponding to the revised version of the manuscript ***

Reviewer #1 (Remarks to the Author):

This paper, evaluating the potential for differential responses in brown trout to temperature exposure based on origin in a cold or warm streams, is of considerable interest for our understanding of thermal influences on physiology, behavior and ecology, as well as the ability for populations from different thermal environments to respond and keep pace with climate change. The manuscript is beautifully written with nice grounding in the literature and theory, well done. Conclusions are generally reasonably caveated based on the limitations of the study (save for the overly strong title, see below), and the authors thoughtfully pointed to where future work could deepen our understanding of the facets that were more limited in providing conclusive evidence given their design; demonstration of genetic distances is a nice foundation. Analyses are sufficient and data clear – but study design needs more clarification/detail within the manuscript. I have limited comments as generally, this is a solid paper that, with appropriate attention to the needs below, will be a nice addition to the literature.

Response #1: Thank you for the kind words and positive overall impression of our manuscript. We appreciate the constructive feedback you have given us below and hope we have satisfactorily addressed all of your points.

Title – I think the title is too strong given the lack of population-level replication in the study, e.g. multiple populations each from a cold and warm environment (which also would help address the confounding between geographic and genetic difference). This is what it is, given the system available, but the conclusions need to match the bounds of the design. Something along the lines of “Increased energetic efficiency observed in brown trout from warm streams versus a cold stream in a northern latitude system” would be more appropriate than the strong causative conclusion conveyed in the current title.

Response #2: We appreciate that the original title may have overgeneralised our results. Reviewer #2 also suggested a new title, so we have gone with something intermediate between the two: “*Chronic exposure to warmer streams increases energetic efficiency of brown trout in a northern latitude system*”.

The journal may limit figure space, but it would be helpful, if at all possible, to include the map and experimental approaches Figure (S1) in the actual paper, in part as the study design is unclear (to me, at least, see below). The figure legend text describing the visually indicated break in distance between IS12 and IS1 is important but confusing as there is no distance legend given to then convey the comparative distance between IS1 and IS5. Suggest adding a distance legend.

Response #3: We only had five display items in the main manuscript (4 figures, 1 table), so we have moved Figure S1 into the main text, citing it as the new Figure 5 at the beginning of the methods. We have also included a distance legend to clarify the distances between the other streams in the map.

[figure redacted]

Ln433: “*Fig. 5. Overview of the Hengill system and experimental approaches. (a) Map of the Hengill geothermal system, indicating the streams from which brown trout and experimental prey were collected, and the streams in which experiments were conducted. Note the distance from IS1 to IS12 is approximately 1 km, but the length of the main river is shortened (as indicated by two diagonal lines on the map) for the purposes of viewing the entire system more effectively. Stream codes are the same as those used in previous publications on the Hengill system^{66,87–89}. (b) An experimental chamber used for in situ metabolic rate measurements, containing a miniDOT logger for monitoring dissolved oxygen concentration and a single individual brown trout. (c) Example of an experimental run for measuring metabolic rates of fish. (d) Example of an experimental run for measuring feeding rates of fish.*”

L375 if energy expenditure/intake is greater than 1, doesn't this mean insufficient energy (<1) to meet demands – i.e. should this ratio be flipped (intake/energy >1)?

Response #4: We apologise for the confusion here by the ordering of our words. However, it should be noted that when we define energetic efficiency at Ln58 and Ln566, it is the ratio of energy intake to expenditure, i.e. values <1 do mean insufficient energy to meet demands. We have rephrased this line to “*the ratio between energy intake and expenditure*”

L429 – first mention of microsatellites as marker used. State in L424 that ___# of microsatellites were genotyped.

Response #5: We now note that 17 microsatellites were genotyped and we refer to a supplementary table with the names of all the loci.

Ln454: “*At the end of all experiments (see below), fin clips were taken for population genetics and preserved in 96% ethanol, with a total of 17 microsatellites genotyped (Table S9).*”

Experimental design needs clearer description. L461 made me wonder why IS12 wasn't included in the experimental locales; assumedly it's because these other sites spanned the broad variation in temperatures you were targeting in your experiment (?), so state that. It would also be good if you could indicate, or color code with a legend, the temperatures at each site on the map. Finally, I don't understand how the fish from the cold stream and warm streams were allocated to the experimental design: L 473 said "up to 10 fish" – does this mean 3 from each focal population + the 1 (empty) control at each site? And what happened when fewer fish were used – from which populations? This all just needs a bit more thought given to the narrative.

Response #6: Yes, the chosen streams evenly spanned the temperature gradient we were targeting in the experiment. We have also clarified that IS12 is further from the other streams and so was not used because it would have involved transporting fish back and forth over much greater distances, which could have become stressful (Ln497). We have added the temperatures of each stream to the map in Figure 5 (see Response #3). We have also specified the number of fish used from each stream in a typical experimental run and that we always ensured a balanced number were used from the cold and warm streams (Ln512).

Ln497: "Note that IS12 was not included in the experimental streams because it is much further from the others, so would have involved transporting fish over greater distances and was not needed to evenly span the temperature gradient."

Ln512: "This typically involved five fish from the cold stream (IS12) and five fish from the warm streams (e.g. two from IS1 and three from IS5, or vice versa). Occasionally, an experimental run had less than ten fish, but we ensured a balanced number of cold and warm origin fish were used in these cases."

L508 IS7 not on map – add.

Response #7: We have now added the label for IS7 to the map in Figure 5 (see Response #3).

Reviewer #2 (Remarks to the Author):

This is an interesting paper taking advantage of a study system of cold and warm streams in Iceland that are at contrasting temperatures due to geothermal warming despite their spatial proximity. The findings are interesting (although I have some concerns about their validity) and the manuscript is mostly well-written. I do have some concerns regarding the methodological and statistical approaches, as well as some suggestions for improving clarity, the presentation of the results, and various other aspects of the manuscript, as detailed below.

Response #8: Thank you for the positive feedback and the constructive comments that have helped to improve our manuscript. We have clarified the methodological and statistical approaches and conducted some additional tests to support our interpretation. Please see our more detailed responses below.

1) There is a potential issue with the approach in that only one cold stream and two warm streams were used. It is therefore difficult to attribute the different pattern in the "cold fish" to the temperature itself – it is hard to establish a causal relationship without

comparing multiple cold and multiple warm populations. Can you eliminate the possibility that there is some other abiotic or biotic factor that differs between these cold and warm streams that could have led to these physiological differences?

Response #9: We acknowledge in the revised manuscript that the gold standard would be to run a reciprocal transplant experiment with multiple cold and warm populations, but this was just not feasible due to the small population size of brown trout in the Hengill system and the lack of similar systems with brown trout for catchment-level replication (Ln427). We stated in the original manuscript that the water chemistry of all streams in the system is very similar, but we have now expanded on that to note that there are narrow ranges of pH, sulphate, and the key nutrients and minerals that are most likely to influence the physiology of the trout (Ln417). We have also included a new supplementary figure, including a statistical comparison of 12 environmental variables that we have measured across multiple years to show that there is no significant difference among the streams (Figure S5). Thus, we can be reasonably confident that temperature is the key environmental variable that differs between these streams (as argued in all our previous papers from the system).

Ln427: *“The gold standard for a reciprocal transplant experiment like this would be to use multiple cold and warm populations, ideally from different river systems, but the small population sizes of brown trout and uniqueness of the Hengill geothermal system limited the current undertaking to these three streams.”*

Ln417: *“Importantly, the indirect nature of this heating means that physical and chemical characteristics are very similar across all streams^{87,89}, with no significant difference among streams in pH, sulphate, and key nutrients and minerals (Fig. S5).”*

Figure S5: “*Similarity in key environmental variables apart from temperature in the three study streams. 12 environmental variables were sampled in the three study streams in August 2004, August 2008, and April 2009 (see references 87 and 89 in the main text for methodological details). There was no significant difference in the concentration of pH ($F_{1,7} = 0.125$, $p = 0.733$), total nitrogen ($F_{1,7} = 0.854$, $p = 0.386$), total phosphorous ($F_{1,7} = 0.960$, $p = 0.360$), ammonia ($F_{1,7} = 0.933$, $p = 0.366$), nitrate ($F_{1,7} = 0.045$, $p = 0.838$), phosphate ($F_{1,7} = 0.255$, $p = 0.629$), sulphate ($F_{1,7} = 0.800$, $p = 0.401$), calcium ($F_{1,7} = 0.425$, $p = 0.535$), potassium ($F_{1,7} = 1.046$, $p = 0.340$), magnesium ($F_{1,7} = 0.008$, $p = 0.933$), sodium ($F_{1,7} = 1.941$, $p = 0.206$), or chlorine ($F_{1,7} = 0.135$, $p = 0.724$) among the three streams.*”

2) I question the appropriateness of the statistical approach – the authors chose to merge the data for the two warm streams. However, related to the point above, I would argue that it is crucial to show whether or not the two separate warm streams both exhibit the same pattern as each other. If this pattern is driven by only one of the warm streams, that calls into question the conclusion that the observed increased energetic efficiency is due to warming as opposed to some other attribute of the environment in one of these warm streams. Indeed, in Figure S5a, it doesn’t look like there is any evidence for increased energetic efficiency in the IS5 population – this seems to only be the case for IS1. Statistical results need to be provided for the main effects and interaction for the relevant supplementary figures for your results to be convincing (in fact, in my opinion, these supplementary figures should be in the main text instead). When looking at these supplementary figures, it also becomes clear that the sample size for IS1 is actually quite small. All sample sizes should be stated clearly in the main text.

Response #10: For some additional context, the trout population in the Hengill system is small and we are always conscious of minimising the impact we may have through studying them. We took the decision at the outset to utilise fish from two warm streams in the system that are quite close geographically and in their temperature regime to ensure sufficient numbers of the size range we were targeting, and compare them to fish from a single cold stream where there were many more individuals to select. Our study design thus reflected this choice as now noted at Ln423. Our genetic analysis later supported the interpretation of the fish from these two warm streams as a putative single population that was genetically distinct from the cold stream and so we prefer to retain the cold v warm figures in the main text with the breakdown by the three streams in the supplementary. We have, however, taken steps to reassure readers that the two warm streams act in a very similar fashion, adding full statistical tables to the supporting information (Tables S2, S5, and S8) and providing the pairwise comparison of slopes for all streams in Figures S3 and S4 (see quotations below). It should be noted that the slopes of the two warm streams are significantly different from the cold stream slope for both feeding rate and energetic efficiency (with one exception for feeding rate) and the slope of the two warm streams are never significantly different from one another. We think this adds strong support for the interpretation that the fish from the warm streams behave in a similar way, but differently to the fish from the cold stream (i.e. patterns are not driven by only one of the warm streams). We have also added sample size information to the manuscript and it should be noted that we used exactly the same number of fish from IS1 and IS5 (Ln420).

Ln423: “*Note that we selected fish from two warm streams that were close to each other geographically and in their temperature regimes to ensure we had sufficient numbers for experiments whilst minimising the impact on these relatively small population sizes. The trout are much more plentiful in IS12 and thus a single cold stream sufficed to source fish for the experiments.*”

New text in Figure S3: “Note that for *Radix balthica*, the slope of IS12 (0.0013 ± 0.0317 ; mean \pm 95% CI) was significantly different from IS1 (0.0677 ± 0.0557 ; $t = 2.39$, $p = 0.020$), but not IS5 (0.0432 ± 0.0544 ; $t = 1.54$, $p = 0.127$), while IS1 was not significantly different from IS5 ($t = -0.77$, $p = 0.443$). For *Simulium vittatum*, the slope of IS12 (-0.0164 ± 0.0602 ; mean \pm 95% CI) was significantly different from both IS1 (0.0932 ± 0.1032 ; $t = 2.13$, $p = 0.037$) and IS5 (0.1068 ± 0.1056 ; $t = 2.33$, $p = 0.022$), while IS1 was not significantly different from IS5 ($t = 0.23$, $p = 0.823$).”

New text in Figure S4: “Note that for *Radix balthica*, the slope of IS12 (-0.3592 ± 0.0999 ; mean \pm 95% CI) was significantly different from both IS1 (0.5395 ± 0.3228 ; $t = 5.57$, $p < 0.001$) and IS5 (0.0850 ± 0.3833 ; $t = 2.32$, $p = 0.023$), while IS1 was not significantly different from IS5 ($t = -1.46$, $p = 0.148$). For *Simulium vittatum*, the slope of IS12 (-0.3825 ± 0.1004 ; mean \pm 95% CI) was significantly different from both IS1 (0.0191 ± 0.3246 ; $t = 2.47$, $p = 0.016$) and IS5 (0.0076 ± 0.3855 ; $t = 2.02$, $p = 0.046$), while IS1 was not significantly different from IS5 ($t = -0.055$, $p = 0.956$).”

Ln420: “A total of 86 brown trout (65–180 mm fork length) were collected from three streams in the system where they are particularly abundant: 44 from a cold stream (IS12 with a mean annual temperature of 7.8 ± 4.2 standard deviations °C); and 42 from two warm streams (21 from IS1 = 11.3 ± 4.0 °C and 21 from IS5 = 13.8 ± 1.6 °C; see Fig. 5a).”

3) Another concern I have is that the authors appear to have omitted mentioning relevant literature, perhaps in an effort to overstate the novelty of this work. They do not cite studies that seem directly relevant to this topic, including some on the metabolic rate of other species in Iceland that are exposed to chronic warming in geothermally heated freshwater systems (e.g. threespine sticklebacks, Pilakouta et al. 2020 Functional Ecology) or species in other locations where there is exposure to chronic warming (e.g. European perch, Sandblom et al. 2016 Nature Communications).

Response #11: We apologise for the oversight. It was not our intention to overstate the novelty of our work, and we are grateful that you have highlighted these two very relevant studies to mitigate that impression. We now acknowledge them explicitly in both the introduction (Ln84) and discussion (Ln282) and have added citations to them as support for several of our statements throughout the paper. Generally though, we feel the 101 references to past research in our original submission was a respectable effort at summarising the extensive literature on this topic and placing our results in that broader context.

Ln84: “Alternatively, organisms could downregulate their metabolism following sustained exposure to warmer environments^{28,29}. For example, wild fish populations that have been exposed to chronic warming for many generations have been shown to exhibit lower basal metabolic rates than ambient populations, which can contribute to faster growth rates^{30,31}.”

Ln282: “It should be noted that the reduced metabolic rate observed in perch and sticklebacks following chronic exposure to warmer environments was only observed for basal metabolic rates and so does not necessarily contradict our findings^{30,31}. Future experiments comparing the thermal sensitivities of metabolic and feeding rates should thus consider basal and maximum metabolic rates, in addition to the routine (field) metabolic rates quantified here.”

4) I believe the presentation of the results could be significantly improved. a) For example, why are three stream names IS12, IS1, and IS5? Can you use something more intuitive for presenting the results for these different streams? On L461, you then mention IS1,

IS5, IS8, IS11, and IS16, which are designated as “experimental streams” in Figure S1, but it is unclear what this means.

Response #12: We now note that we have called the streams IS12, IS1, and IS5 for consistency with the stream coding system used in previous publications on the Hengill system (Ln438). To make them more intuitive, we now specify at key places in the manuscript that IS12 is the cold stream and that IS1 and IS5 are the warm streams, e.g. when first mentioned at the beginning of the results (Ln149) and in the graphical legends of all relevant figures, both in the main text and supporting information (see an example below for Figure 2). We also clarify that the “experimental streams” acted as a sort of natural laboratory for conducting metabolic rate or feeding rate experiments, rather than having to take the fish back to more artificial conditions in a laboratory (Ln493).

Ln438: “Stream codes are the same as those used in previous publications on the Hengill system^{66,87–89}.”

Ln149: “In relative terms, the differentiation was strongest for the IS12-IS1 (cold v warm) and IS12-IS5 (cold v warm) comparisons, and weakest for the IS1-IS5 (warm v warm) comparison (Table 1; see Fig. 1a for a map of the study site).”

Ln493: “Here, these experimental streams acted as a sort of natural laboratory, whereby we could conduct metabolic rate experiments at different temperatures without needing to bring fish back to the laboratory to do so in an artificial setting under temperature-controlled conditions.”

b) The use of colour is inconsistent and confusing. IS1 is shown in orange in Figure 1a but then it’s merged with IS5 and shown in red in Figure 1b.

Response #13: We originally used the orange colour for IS1 in Figure 1a for consistency with the use of orange for that stream in the supplementary figures. We recognise that this could be confusing for readers that focus solely on the main text, however, and so we have used the same red colour for both IS1 and IS5 in Figure 1a, whilst still using different symbols for each stream.

c) Figure 2: The caption should provide some information on which populations/streams the “cold” and “warm” correspond to. Also, why aren’t there separate regression lines shown for cold and warm in these plots? Even if “source stream temperature” doesn’t have a significant effect on metabolic rate, it would be useful to show these separate lines on the plots.

Response #14: For greater clarity, we have included both stream names and specified which are warm or cold in the graphical legends of all relevant figures in the main text and supporting information (see Response #12). We cannot fit separate regression lines for the cold and warm streams in Figure 2 because source-stream temperature was not included in the optimum model for metabolic rate, i.e. the best model indicates that metabolic rate is only dependent on temperature and body mass, not source-stream temperature. We have clarified this in the figure legend.

Ln202: *“Thus, a single regression line is fitted in each panel because source-stream temperature (warm v cold) was not included in the optimum model.”*

5) The Discussion is very focused on brown trout and specifically the Hengill study system with limited effort to broaden the scope and relate this to relevant literature. I suggest reframing some of the Discussion to try to place your results in a broader context.

Response #15: We are not sure this comment is entirely fair. For instance, we compared our findings to those of previous studies on cyprinids (Ln276) and medaka fish (Ln279), as well as kelp, coral, and snails (Ln314). We also have a large paragraph discussing the ecology of the freshwater snail and blackfly larvae prey used in the experiments (Ln337). Nevertheless, we now also compare our findings to past studies on cyprinids and sticklebacks (Ln282; see Response #11).

Consider rephrasing the title: “Chronic exposure to a warmer environment increases energetic efficiency of brown trout”

Response #16: We also had a suggestion from Reviewer #1 to change the title so it is more specific to the study region. Accordingly, we have gone with the title *“Chronic exposure to warmer streams increases energetic efficiency of brown trout in a northern latitude system”*.

L27-28: This statement is not specific to predatory organisms, right? I suggest rewording.

Response #17: We have deleted the word “*predatory*” and now just refer to “*organisms*”.

L27-30: A bit of a jump in logic between sentences here – why does warming lead to an increase in metabolic rates? Need to specify if talking about ectotherms.

Response #18: We have added a new opening sentence to the abstract to eliminate the jump in logic.

Ln25: “*Metabolic rate determines the amount of energy an organism needs to survive, and it is typically predicted to increase with warming up to an optimum temperature for ectothermic organisms.*”

L29: “However” doesn’t fit in the middle of the sentence.

Response #19: Deleted.

L33-34: Specify how many cold and warm streams you sampled.

Response #20: Done.

L92: Avoid using / as it is too informal for a scientific paper.

Response #21: We have rephrased as “*for local or even global extinctions to occur*”

L144-160: This feels really out of place as the last paragraph in the Introduction. In my opinion, it would be more appropriate to move this to the Discussion. This would also help shorten the Introduction, which I found a bit too long.

Response #22: Our logic for placing it at the end of the introduction was to avoid a reader expecting something different from our study and being frustrated that we were not disentangling phenotypic plasticity and genetic adaptation in our results section. We agree that it is an unorthodox way to end an introduction though and have moved it to the second last paragraph of discussion, where we feel it works well as a closing caveat for our study.

I suggest avoiding abbreviations such as MTE which require the reader to remember what these refer to, but it is ultimately up to the authors to decide whether to keep this.

Response #23: In this instance, we prefer to retain the abbreviation MTE because it is quite an established acronym, and Metabolic Theory of Ecology is frequently referred to as MTE in other papers.

*** Note that original reviewers' comments are in bold. Our responses are numbered sequentially, written in plain font, and contain quotations to the revised text in italics with line numbers corresponding to the revised version of the manuscript ***

Reviewer #1 (Remarks to the Author):

I reviewed the earlier draft of this manuscript which remains well-written and elegantly couched in the literature. I apologize if my original review did not sufficiently emphasize concern about the overly strong conclusions given the minimal study design (which is not by fault of the authors, just a constraint of the setting, with insufficient replication and very small sample sizes), but I'm afraid those concerns still stand. This paper provides a well-crafted theoretical framing around a question of true importance, but one for which the study limitations simply do not align with that lofty setting or enable such firm conclusions. These concerns are elevated at this point given the high caliber of this particular journal, which notes in its guidelines "There should be a discernible reason why the work deserves the visibility of publication in a Nature Portfolio journal rather than the best of the specialist journals." This is of course an editorial decision, but my recommendation would be that the paper merits publication after further (relatively easy) strengthening of its narrative description of the study design and caveated interpretation of the results (there are slight improvements in this revision but still work to be done in my opinion) – but in a different, more specialized journal. Mainly, I feel the conclusions are still too strong given the lack of replication. Also, the expectations for interpreting the genetic data are not appropriately caveated (except for in the discussion) given the confounding influence of isolation by distance and small population sizes on genetic outcomes (I should have commented on this more deeply in my original review so again, I apologize for the rather light input originally). Given the organization of this journal (introductions, results, discussion, methods), my comments are a bit interspersed, but I hope they are helpful in a next revision for hopeful publication in another journal..

Response #1: We appreciate the positive feedback on the construction of our manuscript, but of course are disappointed that you remain sceptical about the suitability of the paper for *Communications Biology*. We are grateful for the opportunity from the editor to respond to your outstanding concerns and have done our best to take on board your feedback, particularly through greater caveats around interpretation of the results and reducing the prominence of the genetic component of the study. We hope these changes improve your confidence in our study.

I appreciate the revision of the title to narrow the findings to a given system, but it is still too causative and not appropriately caveated given the limited study design – it should emphasize that increased energetic efficiency was observed (correlated) in this system, not caused by exposure to warmer temperatures.

Response #2: We have now changed this to a more neutral title, which does not make claims of causality: "*Fish originating from warmer streams in a northern latitude system exhibit increased energetic efficiency*".

138: The authors present nice coverage of the limitations of their genetic data in the Discussion, but the set up of genetic outcomes "as a function of the thermal origin" in this hypothesis is one thing that gives me greater pause on 2nd review as, especially with the lack of replication, there is no way to separate the influence of isolation by temperature from isolation by distance (as noted in the Discussion). Additionally, a point not covered in the paper at all is that in small populations like with assumedly small effective sizes,

the degree of genetic differentiation observed between the cold and warm streams (without replication) could be entirely explainable by drift as this influence is generally greater in small populations than selection.

Response #3: We understand your concerns here and have decided to restructure the manuscript accordingly. We now put the ecophysiology results first and the population genetics results second to give greater emphasis to what we believe is our strongest finding, i.e. fish from warm source streams exhibit greater energetic efficiency with increasing experimental temperature than fish from the cold source stream. We now pitch the population genetics analyses as only providing additional context to aid interpretation of the ecophysiology results in the revised introduction (Ln139). Our population genetics results simply show that the cold stream forms a distinct genetic cluster from the two warm streams. You note that this pattern could have resulted by chance via genetic drift, but we examine putatively neutral markers, so the observed differentiation must reflect some balance between drift and gene flow. The question really is whether any gene flow is fully symmetrical among all possible pairs of study populations, or actually is more restricted between the cold population and the two warm populations (either owing to simple geography, or isolation-by-distance reinforced by isolation-by-environment). Our findings point towards the latter, given that the cold stream is much more different genetically from both the warm streams than they are from each other. Whilst we cannot rule out a scenario of “drift with equal gene flow”, this seems less parsimonious to us than “drift with unequal gene flow”, given the spatial and environmental configuration of our sampling design. Nevertheless, we now acknowledge that the “drift with equal gene flow” hypothesis is also possible. Moreover (as per the previous submission), we fully acknowledge with cautious language that we cannot, without additional replication at the population level, cleanly distinguish between the isolation-by-distance and isolation-by-thermal-adaptation hypotheses. We simply make the point that, if anything, these two mechanisms are likely to reinforce each other, given that the two warm study populations are closer to each other in space than either is to the cold study population (Ln360).

Ln139: *“Our second aim was to characterise population structure using neutral microsatellite markers, to provide additional context and indirect information on putative patterns of dispersal.”*

Ln360: *“Collectively, these results suggest two genetic units in our dataset, corresponding to a “warm cluster” and a “cold cluster” (Fig. 5). These patterns could, in theory, stem entirely from random genetic drift, where allele frequencies in the cold source stream by chance drifted apart from those in the two warm streams under a situation of fully symmetric gene flow. Indeed, effective population sizes are likely to be low in these small streams. A more parsimonious explanation in our view, however, is that both isolation-by-distance and isolation-by-environment might be going on in this system, and indeed potentially reinforcing one another. That is, gene flow is likely to be constrained between IS12 (cold) and the other two source streams (IS1 and IS5; warm) for both geographic reasons (IS1 and IS5 are closer to each other than either is to IS12) and because immigrants originating from a different thermal regime might be less successful at passing on their genes in the recipient population, thus amplifying genome-wide divergence driven by genetic drift⁴⁹.”*

164: Even though 2 clusters may be merited, all of these results do not suggest strong affinity for the two clusters by stream thermal profile (i.e., a cold vs hot cluster). They are generally as expected with isolation by distance, with IS1 falling genetically in between the other two streams in line with its being between them geographically.

Response #4: Yes indeed, and we fully acknowledge in the revised manuscript that this could all be explained by simple isolation by distance. And that any additional isolation-by-thermal-adaptation would tend to reinforce, rather than counteract, these geographic effects.

Ln373: *“With our current study design, we are unable to cleanly separate isolation-by-distance versus isolation-by-adaptation as drivers of neutral genetic divergence among trout originating from warm versus cold streams because temperature differences among the studied streams are confounded with river distance. Local thermal adaptation correlating with population genetic structure has been reported in numerous other systems^{65,66} (but see^{67,68}), including for *Radix balthica* in distinct geothermal habitats in Iceland⁴⁸. It is thus possible that the contrasting thermal sensitivities that we observed for feeding rate among fish originating from cold versus warm source populations may be due to local adaptation at a genetic level, but this could also reflect non-adaptive genetic divergence, or purely environmental effects on phenotypes (flexible physiological remodelling). Phenotypic plasticity, in turn, could operate within generations (e.g. early-life acclimation⁶⁹) or across generations (e.g. parental effects, epigenetic inheritance^{70,71}), and be adaptive, maladaptive or neutral with respect to fitness. Additional experimental work, ideally coupled with functional genetics, would be required to distinguish among these various possibilities, but collectively this highlights the difficulty in anticipating how organisms will respond to warming without incorporating intraspecific variation and its associated eco-evolutionary drivers into predictive frameworks such as MTE.”*

261: Use of the term “chronic exposure” made me think this was referring to (longer-term) experimental exposure, but you mean this based as a descriptor of where the fish originated – clarify this: “chronic exposure due to originating in a population from a warmer stream”

Response #5: We can see how this could be confusing, so we have removed most mentions of chronic exposure from the manuscript, including the title. The sentence you flagged has also been reworded to make it much clearer that we are talking about pre-existing phenotypic differences as a function of stream origin, which then affect how individual fish respond to acute (experimentally induced) changes in temperature.

Ln297: *“In this study, we hypothesised that fish originating from warmer streams would exhibit reduced thermal sensitivity of metabolic rate relative to feeding rate, increasing their energetic efficiency at higher temperatures, as an adaptive response to a history of chronic exposure to consistently higher average temperatures.”*

L283: this specific statement about perch and sticklebacks (“the reduced metabolic rates observed”) makes it seem like you’ve brought this study up before, rather than that you are extending your general discussion based on varied literature including for these fish. Also, the use of “following chronic exposure” here, rather than “for fish originating from populations from different thermal regimes” begs my point above on the use of “chronic exposure”.

Response #6: Good point, we have revised these sentences.

Ln325: *“Reduced basal metabolic rates have been observed in stickleback³⁰ and perch³¹ populations originating from warmer thermal regimes, but this does not necessarily contradict our findings as our study examined routine (field), rather than basal, metabolic rates. Future*

experiments comparing the thermal sensitivities of metabolic and feeding rates should thus consider basal and maximum metabolic rates, in addition to routine metabolic rates.”

293: These kind of statements and the strong conclusions – still – from your title are making me evaluate this paper more critically. This is not founded based on this minimal study design without replication: “This indicates a divergence in thermal responses across organismal traits”

Response #7: This section, and indeed the paper as a whole, has been substantially revised to fully take on board these comments and the limitations of our study design. Note that by “thermal response” here, we mean how individuals from a given source population respond to acute warming (as imposed by transplanting them to a range of destination streams that differ in temperature). The “divergence” bit referred to the fact that the strength of the relationship between feeding rate/energetic efficiency and experimental (destination) temperature varied depending on which origin stream (background thermal regime) the fish were sourced from. In any case, the language was (and remains) careful and cautious – e.g. “suggests”, “may be attributable to”, etc.

Ln336: *“This indicates a divergence in thermal responses across organismal traits, such that fish originating from colder regimes may be less able to increase their feeding rate to match their metabolic demands after acute exposure to warmer environments, leading to a reduction in energetic efficiency and thus potential starvation with warming²⁰. In contrast, fish originating from warmer regimes may be better able to increase their feeding rate to obtain an increased energetic efficiency with acute warming, highlighting the potential for thermal adaptation to increase population persistence in the face of warming. It must be stressed, however, that we cannot conclude from the current study that trout were genetically adapted to the local thermal regimes of their origin stream. Any persistent differences among these putative populations could instead reflect non-adaptive genetic, or else purely environmental divergence (phenotypic plasticity).”*

312: “Collectively, these results suggest two genetic units in our dataset...” – this is true, but the following statement is not grounded given the potentially confounding effect of IBD and drift – “...which may be adapted to a normal ambient stream environment for Iceland and one which is heated by 3.5-6 °C”. There is a good, valid discussion of this following shortly, L 321, that could be better integrated/caveated in these earlier firm statements.

Response #8: We have revised this section as recommended (Ln360; see Response #3 and Ln373; see Response #4).

L364: This idea (that increasing energetic efficiency may contribute to this observation) is interesting and founded as it is posed. It is appropriately caveated with “could”, but note the following statement is not appropriately caveated based on your results: “underpinned by a genetic component that favours warm populations”

Response #9: We have rephrased this sentence in the revision to acknowledge that either genetic adaptation or phenotypic plasticity could be at play.

Ln434: *“Our results indicate that greater energetic efficiency, underpinned by either local adaptation or physiological acclimation, could also play an important role in supporting the higher biomass of trout in the warmer streams.”*

L378: This section appropriately grounds/caveats your interpretation of/goals for the genetic evaluation, in simply gaging if the “scope for phenotypic divergence to be possibly present”. Need to do a better job of adhering to this interpretation in presenting your hypotheses and results.

Response #10: We have done our best now to revise the entire manuscript along these lines, to better adhere to this interpretation in presenting the hypotheses, results, and discussion (see Responses #3 and #4 in particular).

L387: Related to adaptive divergence, given how small you indicated these populations likely are, their effective population sizes may be too low to allow for adaptive divergence. Rather, this may be more likely to be a drift-dominated system, genetically. Add discussion of this.

Response #11: This is now added (Ln360; see Response #3).

L415: I am realizing on this 2nd review that the overarching study manipulation/design was never introduced before going into details. I think the experimental vs study streams aspect to the design needs a clearer, up front narrative explanation that consistently distinguishes *fish* from study populations from different thermal environments and their use in a study design vs the *thermal habitats* of streams used for the in-situ experiment. This description as is gets very into the weeds quickly, and essential elements of this approach are presented piecemeal (L420, L497) without setting up the broader vision. It would benefit from a brief overview description along the lines of: “Our study anchored around three populations of brown trout from different thermal habitats; our experiment consisted of using fish from these 3 study streams in a designed assessment of their metabolic and feeding responses while temporarily exposed to different temperatures in several test streams (or stream reaches of their origin stream) representing the broad thermal variation observed in the system”. This will provide the overview above the subheadings, as a basic study design (for transfer of fish from study populations to thermal experimental streams) applicable to all the manipulations (oxygen consumption, feeding rate, etc). Perhaps describe this under a subheading of General study design.

Response #12: Apologies if this was not clear enough before. We have now included a short paragraph along these lines at the very end of the Introduction section, i.e. to ensure all readers are briefed on the study design before reading the results, even if they do not switch down to read the Methods first. The new paragraph briefly summarises the general study design and very carefully distinguishes between the source streams from which the fish were collected (which differed in thermal regime), and the destination streams to which they were transplanted experimentally in order to measure their acute physiological responsiveness to a range of temperatures. We are also much more careful now throughout the entire revised manuscript to clearly distinguish between source thermal regime (population background) versus current temperature (at the time of physiological measurements) in the destination streams.

Ln151: “Our experimental design entailed collecting fish from three different source streams (one cold: IS12; two warm: IS1 and IS5) and then transplanting them temporarily to a set of different “experimental” streams in the same catchment, which themselves varied naturally in thermal regimes (Fig. 1). In this way, each source population was artificially exposed to the same thermal gradient (range of experimental temperatures), exploiting natural spatial variation in temperatures in the system and thereby emulating the effects of acute warming in

a space-for-time substitution. For logistical and ethical reasons, each fish from a given source stream was transplanted to only a single destination stream, rather than sequentially moving the same individual to all destinations and their associated temperatures (which would introduce order effects and potentially incur excessive stress). Thus, whilst we could not measure within-individual sensitivity of routine metabolic (and feeding) rate to acute warming, we could measure population-level sensitivity. Given that individuals were randomly allocated to experimental streams, any variation in thermal responsiveness among fish originating from the same source stream should reflect acute effects of temperature on physiology, whilst any differences among fish from different source streams should reflect pre-existing (persistent) variation due to inherited (genetic or epigenetic) factors or prior (early-life) acclimation to developmental temperatures experienced in the home stream.”

433: Add “prior to translocation to experimental streams for study, trout were maintained...”

Response #13: Now added. However, we have used the word transplantation instead of translocation to be consistent with the terminology used in the study.

Ln486: “Prior to transplantation to the experimental streams the trout were maintained in their home streams in cylindrical white plastic arenas (250 mm diameter, 300 mm height).”

L453: I suggest presenting the genetic methods at the end, as it is confusing here without the clearer study design laid out – and you note the collections were done after the experiments. How you treated fish from these populations (by merging IS1 and 5) given the genetic outcomes can be presented more up-front in the results, but I think should come last in the methods.

Response #14: We have now reorganised the entire manuscript such that the genetic aspects follow the experimental physiology aspects, in both the methods and the results.

473: Suggest strengthening the recognition of the distance/temperature conundrum by removing the word “with” after “network,” and adding “meaning that across the isolation by distance template there are” in this phrase: “...the same river network, with no physical longitudinal barriers or confounding environmental factors other than temperature”. Again, it is nicely stated in Discussion, L260: “consequence of isolation-by-distance and/or isolation-by environment, which was supported by our genetic results” but this limitation needs to be presented here given that distinguishing these two is important for your ultimate conclusions and yet you cannot distinguish them at all.

Response #15: Revised as suggested.

493: Try to strengthen the clarity around study fish vs experimental streams wherever you can, here, e.g., by adding after “measured” the descriptor “for study stream fish in the experimental design carried out”

Response #16: We have done our best in the revised manuscript to be clearer about stream origin vs experimental stream throughout, and have made some specific changes to this section.

Ln499: “Here, these experimental streams acted as a sort of natural laboratory, whereby we could conduct metabolic rate experiments at different temperatures (in the experimental streams to which fish from the three source streams were transplanted) without needing to

bring fish back to the laboratory to do so in an artificial setting under temperature-controlled conditions. These experimental streams were chosen to best span the range of available temperatures during sampling (4.6 to 19.7 °C)."

498: Add further clarity along these lines: "Note that, though fish from IS12 were included as a study population, IS12 itself was not included as an experimental stream..."

Response #17: We have clarified things here according to your suggestion.

Ln505: *"Note that although fish from IS12 (cold source stream) were included as one of the three study populations, IS12 itself was not included as an experimental stream because it is much further from the others, so would have involved transporting fish over greater distances and was not needed to evenly span the temperature gradient"*

506: Similarly, revise to read "A single brown trout individual from a given study population was placed inside..."

Response #18: Done.

511: Similarly, revise to read "Up to ten chambers containing fish were placed in each thermally-targeted experimental stream for each experimental run,..."

Response #19: Done.

512: This may be somewhat redundant with the comment above but good added detail of the type that is more generally needed (first tangible mention of study fish being used across this design). I would move it, and the following sentence (Note that...) up as part of a general description of the overall experimental manipulation/design before the description of how you did the measurements: "This typically involved five fish from the cold stream (IS12) and five fish from the warm streams (e.g. two from IS1 and three from IS5, or vice versa)." Then you could follow with the "Note that" statement about not including IS12 as an experimental stream, followed by the other "Note that" statement, edited as needed for transition, mentioning the acclimatization of fish in the experimental streams before measurements were taken.

Response #20: We are clearer about stream origin vs experimental stream throughout, and have made some specific changes to this section and elsewhere accordingly.

547: Revise to read "in the same thermally-targeted experimental stream"

Response #21: Done.

564: Note to editor, this is not at all my realm of expertise so I am not positioned to evaluate the sufficiency of this approach.

Response #22: Our approach follows methodology established by Vucic-Pestic *et al.* (2011) in *Global Change Biology* and should be a robust estimation of energetic efficiency.

565: This is a good descriptor the type of which could, aligning with my general recommendation for more consistent clarity on design, be included to ground your hypothesis for each subheading/experiment; but suggest revising further to read: "To

determine the potential impact of source-stream temperature on the energetic constraints of brown trout evaluated across different experimental temperatures...”

Response #23: Done.